# The sensitivity of $pCO_2$ reconstructions to sampling scales across a Southern Ocean sub-domain: a semi-idealized ocean sampling simulation approach

Laique M. Djeutchouang[1,2], Nicolette Chang[1], Luke Gregor[3], Marcello Vichi[2], Pedro M. S. Monteiro[1]

[1]SOCCO, CSIR, Rosebank, Cape Town, 7700, South Africa
[2]MARIS, Department of Oceanography, University of Cape Town, Cape Town, 7701, South Africa
[3]Environmental Physics, Institute of Biogeochemistry and Pollutant Dynamics, ETH Zürich, Zürich, 8092, Switzerland

*Correspondence to*: Laique M. Djeutchouang (merlindjeutchouang@gmail.com)

**Abstract.** The Southern Ocean is a complex system yet is sparsely sampled in both space and time. These factors raise questions about the confidence in present sampling strategies and associated machine learning (ML) reconstructions. Previous studies have not yielded a clear understanding of the origin of uncertainties and biases for the reconstructions of the partial pressure of carbon dioxide ($pCO_2$) at the surface ocean ($pCO_2^{ocean}$). We examine these questions through a series of semi-idealized observing system simulation experiments (OSSEs) using a high-resolution ($\pm10$km) coupled physical and biogeochemical model (NEMO-PISCES). Here we choose one year of the model sub-domain of 10 degrees of latitude (40ºS – 50ºS) by 20 degrees of longitude (10ºW – 10ºE). This domain is crossed by the Sub-Antarctic Front and thus includes both the Sub-Antarctic Zone and Polar Frontal Zone in the South-East Atlantic Ocean, which are the two most sampled sub-regions of the Southern Ocean. We show that while this sub-domain is small relative to the Southern Ocean scales, it is representative of the scales of variability we aim to examine. The OSSEs simulated the observational scales of $pCO_2^{ocean}$ in ways that are comparable to existing ocean $CO_2$ observing platforms (Ships, Wavegliders, Carbon-floats, Saildrones) in terms of their temporal sampling scales, and not necessarily their spatial ones. The $pCO_2$ reconstructions were done using a two-member ensemble approach that consisted of two machine learning (ML) methods, (1) the feed-forward neural network and (2) the gradient boosting machines. The baseline data were from the ship-based simulations mimicking ship-based observations from the Surface Ocean $CO_2$ Atlas (SOCAT). For each of the scale-sampling scenarios, we applied the two-member ensemble method to reconstruct the full sub-domain $pCO_2^{ocean}$. The reconstruction skill was then assessed through a statistical comparison of reconstructed $pCO_2^{ocean}$ and model domain mean. The analysis shows that uncertainties and biases for $pCO_2^{ocean}$ reconstructions are very sensitive to both the spatial and temporal scales of $pCO_2$ sampling in the model domain. The four key findings from our investigation are: (1) improving ML-based $pCO_2$ reconstructions in the Southern Ocean requires simultaneous high-resolution observations (< 3 days) of the seasonal cycle of the meridional gradients of $pCO_2^{ocean}$; (2) Saildrones stand out as the optimal platforms to simultaneously address these requirements; (3) Wavegliders with hourly/daily resolution in pseudo-mooring mode improve on Carbon-floats (10-day period), which suggests that sampling aliases from the 10-day sampling period might have a  greater negative impact on their uncertainties, biases, and reconstruction means; and (4)

the present seasonal sampling biases (towards summer) in SOCAT data in the Southern Ocean may be behind a significant winter bias in the reconstructed seasonal cycle of $pCO_2^{ocean}$.

## 1 Introduction

The Southern Ocean (SO) remains the world's largest modulator for the ocean uptake of anthropogenic $CO_2$ (Sabine et al., 2004; Frolicher et al., 2015; Friedlingstein et al., 2020). Therefore, reducing uncertainties and biases in $CO_2$ budget estimates in the region is important to better assess and understand its influence on regional and global climate (Majkut et al. 2014; Gruber et al., 2019; Hauck et al., 2020). For instance, since the early 2000s, the SO carbon sink has undergone a reinvigoration characterized by a substantial strengthening as reported by (Landschützer et al., 2015), following a decade (the 1990s) of weakening trends (Canadell et al., 2021; Le Quéré et al., 2007). Based on these findings, many studies have been conducted recently to investigate what drives these inter-annual and decadal changes in the SO carbon sink and assess the uncertainties of the estimates (Bushinsky et al., 2019; DeVries et al., 2017; Fay et al., 2018; Gregor et al., 2018, 2019; Landschützer et al., 2016; McKinley et al., 2020). However, there have not been many studies looking into the role of intra-seasonal and seasonal modes of variability on the uncertainties and biases reported in empirical $CO_2$ mapping approaches (Landschutzer et al., 2016; Gregor et al., 2019). In this region, surface ocean $CO_2$ observations underlying $CO_2$ reconstructions are very sparse, especially during the stormy autumn and winter seasons, requiring a substantial number of extrapolations to map and subsequently fill the gaps due to data spareness (Gregor et al., 2017, 2019; Landschützer et al., 2014).

Many empirical approaches such as statistical interpolations and regression methods (Iida et al., 2015; Jones et al., 2015; Rödenbeck et al., 2014) gained attention as alternative methods to ocean biogeochemical models (Lenton et al., 2013) until recently when Machine Learning (ML) approaches have been used increasingly as an alternative (Denvil-Sommer et al., 2019; Gregor et al., 2017, 2019; Landschützer et al., 2013, 2014, 2016). These novel mapping methods all seek to fill the spatial and temporal sampling gaps from existing ship-based surface ocean $CO_2$ observations by extrapolating the $CO_2$ partial pressure ($pCO_2$) at the surface ocean ($pCO_2^{ocean}$) using prognostic proxy variables (such as satellite-observed and re-analysis-based sea surface temperature, sea surface salinity, mixed layer depth, chlorophyll-a, etc.). The feasibility of these extrapolations is justified through the non-linear relationships between the surface ocean $pCO_2$ and the above-mentioned prognostic variables that may drive changes in the surface ocean $pCO_2$ (Takahashi et al., 1993).

Historically, surface ocean $CO_2$ observations were primarily from voluntary observing ships including research and commercial vessels (Bakker et al., 2012; Pfeil et al., 2013). These $pCO_2$ observations are thus intrinsically biased by the sampling limitations in space and time for the past several decades covering only ~2% of all the monthly 1º observational grid points (Bakker et al., 2016; Sabine et al., 2013). Mainly due to its remoteness and harsh weather especially during stormy

autumn and winter, it has been increasingly shown that the SO is the ocean region that contributes the most to these uncertainties in the contemporary estimates of the mean annual $CO_2$ uptake (Bushinsky et al., 2019; Gloege et al., 2021; Gregor et al., 2019; Ritter et al., 2017). For instance, sparse observations in largely inaccessible SO areas, particularly during the stormy wintertime, have been the biggest barrier to constraining the seasonal cycle of regional and global contemporary ocean-atmosphere $CO_2$ exchange (Bakker et al., 2016; Monteiro et al., 2015; Ritter et al., 2017; Rödenbeck et al., 2015).

Complementary to the increasing effort in the shipboard $CO_2$ observations through the SOCAT initiative, the ongoing development of autonomous ocean observing systems, such as biogeochemical floats and Wavegliders, has started to significantly improve the spatial and temporal coverage of $CO_2$ samples in the SO in recent years (Bakker et al., 2016; Bushinsky et al., 2019; Gray et al., 2018; Monteiro et al., 2015). Over the last decade, the advent of a range of new autonomous ocean observing platforms has opened doors toward closing the seasonal and intra-seasonal sampling biases created by the high cost of ship operations in the Southern Ocean outside the summer window (Bushinsky et al., 2019; Gray et al., 2018; Majkut et al., 2014; Monteiro et al., 2015; Sutton et al., 2021; Williams et al., 2017).

Thus resolving the mean seasonal cycle and intra-seasonal mode of variability through *in situ* observations not only is a challenging exercise but also has followed several avenues from extrapolating findings from the Drake Passage Time-series (DPT) like in (Fay et al., 2018) to utilizing measurements from extended deployments of autonomous ocean observing platforms such as Wavegliders (Monteiro et al., 2015; Nicholson et al., 2022), biogeochemical Argo floats (Bushinsky et al., 2019; Gray et al., 2018; Williams et al., 2017), and more recently using Saildrones (Sutton et al., 2021). These advances have allowed the density of the Southern Ocean surface $CO_2$ observing networks to increase, particularly in the Sub-Antarctic Zone (SAZ) and Polar Frontal Zone (PFZ) which to date are the most observed sub-regions of the SO. Consequently, the problem of general sparseness in observations and particularly the sampling biases (Gloege et al., 2021; Monteiro et al., 2015) was partially addressed but not resolved by the ocean $CO_2$ *in-situ* observations community (Bushinsky et al., 2019; Sutton et al., 2021). For example, under-sampling in winter by ships has been addressed by the 10-day resolution SOCCOM profiling floats and/or pseudo-Lagrangian platforms that are carried zonally by water currents (Bushinsky et al., 2019; Gray et al., 2018; Majkut et al., 2014; Monteiro et al., 2015; Sutton et al., 2021; Williams et al., 2017). (Williams et al., 2017), and then (Gray et al., 2018) reported on persistent differences found with previous $pCO_2$ estimates when the ship-based sampling is sparse, especially during winter, though a recent study seems to disagree on the persistence of these differences (Bushinsky et al., 2019). Therefore, an increase in winter sampling would yield a reduction in the uncertainty levels of surface ocean $pCO_2$ estimates (Bushinsky et al., 2019; Gregor et al., 2019). Notwithstanding these new platforms, sparse and scale-sensitive observations in the Southern Ocean continue to be a barrier to constraining the seasonal cycle and inter-annual variability of surface ocean $pCO_2$ (Monteiro et al., 2015; Rödenbeck et al., 2015; Sutton et al., 2021).

However, we appear to have reached a limit in terms of improving the uncertainties and biases underlying $pCO_2$ reconstructions as reported by (Gregor et al., 2019). According to the authors, the performance measures in existing empirical methods converge, which led the authors to the rhetorical question "have we hit the wall?" In practice, high-quality *in-situ* CO₂ observations like those annually collected and compiled within the SOCAT database (primarily from ships) are fundamental to novel machine learning (ML) methods (Bakker et al., 2016; Sabine et al., 2013), despite the reconstructions being limited by spatial and temporal observational gaps and biased sampling (Gregor et al., 2019). As a result, our understanding of the derived impacts of the Southern Ocean dynamics, particularly seasonal and intra-seasonal modes of variability has remained comparatively poor (Gruber et al., 2019), which may have also been contributing to errors in the $pCO_2$ estimates. At a global scale, Gloege et al. (2021) coupled an observing system simulation experiment (OSSE) with Earth system models to quantify errors in observation-based reconstructions of air-sea CO₂ exchange by using one of the current gap-filling techniques, the self-organizing map feed-forward neural-network (SOM-FFN) by Landschützer et al. (2016). The authors found that errors were regionally high in the Southern Hemisphere, particularly in the SO for which insufficient sampling led to a 31% (15%-58%) over-estimation of decadal variability, but they did not discuss the perspective of uncertainties and biases due to intra-seasonal mode of variability.

This study aims to investigate the sensitivity of the $pCO_2$ reconstructions to the spatio-temporal sampling scales of surface ocean CO₂ observing systems under the assumption that intra-seasonal modes of variability are critical to addressing reconstruction uncertainties and biases. To do that, we used a one-year high-resolution (±10km) coupled physical and biogeochemical forced ocean model for a Southern Ocean sub-region that represents the scales of variability that we aim to resolve. Then, we conduct a series of semi-idealized OSSEs based on existing CO₂ observing platforms (Ships, Wavegliders, Carbon-floats, Saildrones) and coupled with an ensemble of two state-of-the-art machine learning techniques (ML2). A rigorous assessment of the experiment scenarios is conducted through testing and understanding of the ML2 capabilities. We explore the question set by (Gregor et al., 2019) about the prediction uncertainties and biases in contemporary $pCO_2$ reconstructions being now constrained by the sampling scales achievable by the existing ocean observing platforms. We make proposals toward significantly advancing machine-learning reconstructions "beyond the wall". The goal is to find out how the ocean carbon cycle community can better supplement ship-based observations, essential to $pCO_2$ reconstructions, with autonomous platform samples in order to reduce the uncertainties and biases of machine-learning-based mapping approaches.

## 2 Materials and methods

### 2.1 Data source

The data used in this study is from a year-long period of high-resolution (±10km) ocean model simulations. This ocean model is a regional configuration (BIOPERIANT12-CNCRUN05A-S) of the state-of-the-art ocean modelling framework NEMO

(Nucleus for European Modelling Ocean) coupled with the biogeochemical model PISCES (Pelagic Interactions Scheme for Carbon and Ecosystem Studies) which simulates the lower trophic level of the marine ecosystem and the biogeochemical cycles of carbon and nutrients (Aumont et al., 2015). More specifically, we used (1/12)º by (1/12)º-daily simulations of a forced NEMO-PISCES regional Southern Ocean model called BIOPERIANT12 (BP12). There are many prognostic variables including two phytoplankton compartments (diatoms and nanophytoplankton) and a description of the carbonate chemistry in the model. However, we focused only on the variables of particular interest for our study; these variables are the coordinates (time, latitude, longitude), the $CO_2$ partial pressure ($pCO_2$) at the surface ocean ($pCO_2^{ocean}$) and its well-known drivers (Takahashi et al., 1993): sea surface temperature (SST), sea surface salinity (SSS), mixed layer depth (MLD), chlorophyll-a (Chl-a). Their characterization is presented with more details in Table S1.

## 2.2 Data processing and derived variables

In preparation for the training and validation phases of the machine learning (ML) algorithms, some of the input data are transformed for better interpretation. At first, this includes the mixed layer depth (MLD) and chlorophyll-a (Chl-a) data that undergo a log10 transformation to return a distribution closer to a normal distribution (Holte et al., 2017; Maritorena et al., 2010). In practice, existing reconstruction methods have been using MLD climatology as a proxy variable (Gregor et al., 2019; Gloege et al., 2021). This enables a smoothing of the data and thus reduces the uncertainty from MLD information. Therefore, here, using MLD from the model rather than a climatology is likely an advantage compared to the existing methods that use MLD climatology. The advantage of including proxy variables such as MLD and Chl-a is that the model is providing constraints which might not be available from real-world observations. Secondly, it is substantially beneficial to include only the temporal coordinate (time) as a proxy of $pCO_2^{ocean}$. This is because of the characteristics of our study area (Fig. 1a) as being a single domain with no regional or clustering subsets, otherwise, clustering subsets would be used to overcome the spatial limitations that observations present (Gregor et al., 2019). Thus, spatial coordinates (latitude, longitude) are not included in the $pCO_2$ predictors like in (Gregor et al., 2017, 2018) and many other studies used in (Rödenbeck et al., 2015). However, it's important to note that coordinate variables do not drive mechanistic changes in $pCO_2^{ocean}$ according to (Gregor et al., 2017).

The inclusion of the time coordinate as a proxy of $pCO_2$ was done through a variable transformation that aims to preserve the seasonality of the data. More precisely, the preservation of this seasonality is done by transforming the day-of-the-year (j) as in (Gregor et al., 2017); that is,

$$J = \left( cos\left( j.\frac{2\pi}{365} \right),\ sin\left( j.\frac{2\pi}{365} \right) \right) \tag{1}$$

## 2.3 Experimental configurations

### 2.3.1 Study region and selection of the experimental domain

The seasonal cycle is known not only as being the strongest mode of natural variability of $CO_2$ but also the one that most strongly links climate and ocean ecosystems (Mongwe et al., 2018). Given its characteristics that are largely shaped by higher frequencies such as the intra-seasonal mode of variability defining the response modes in physics and biogeochemistry components, the Southern Ocean Seasonal Cycle Experiment (SOSCEx) project was created (cf. Sect. S1.1 for more details). As schematically depicted in Fig. S1, the novel aspect of the third phase of SOSCEx was the integration of a multi-platform

approach that consisted of combining gliders, ships, floats, satellites, and prognostic models to explore new questions about the climate sensitivity of $CO_2$ and ocean ecosystem dynamics and how these processes are parameterized in forced ocean models such as the NEMO-PISCES regional configuration, BIOPERIANT12.

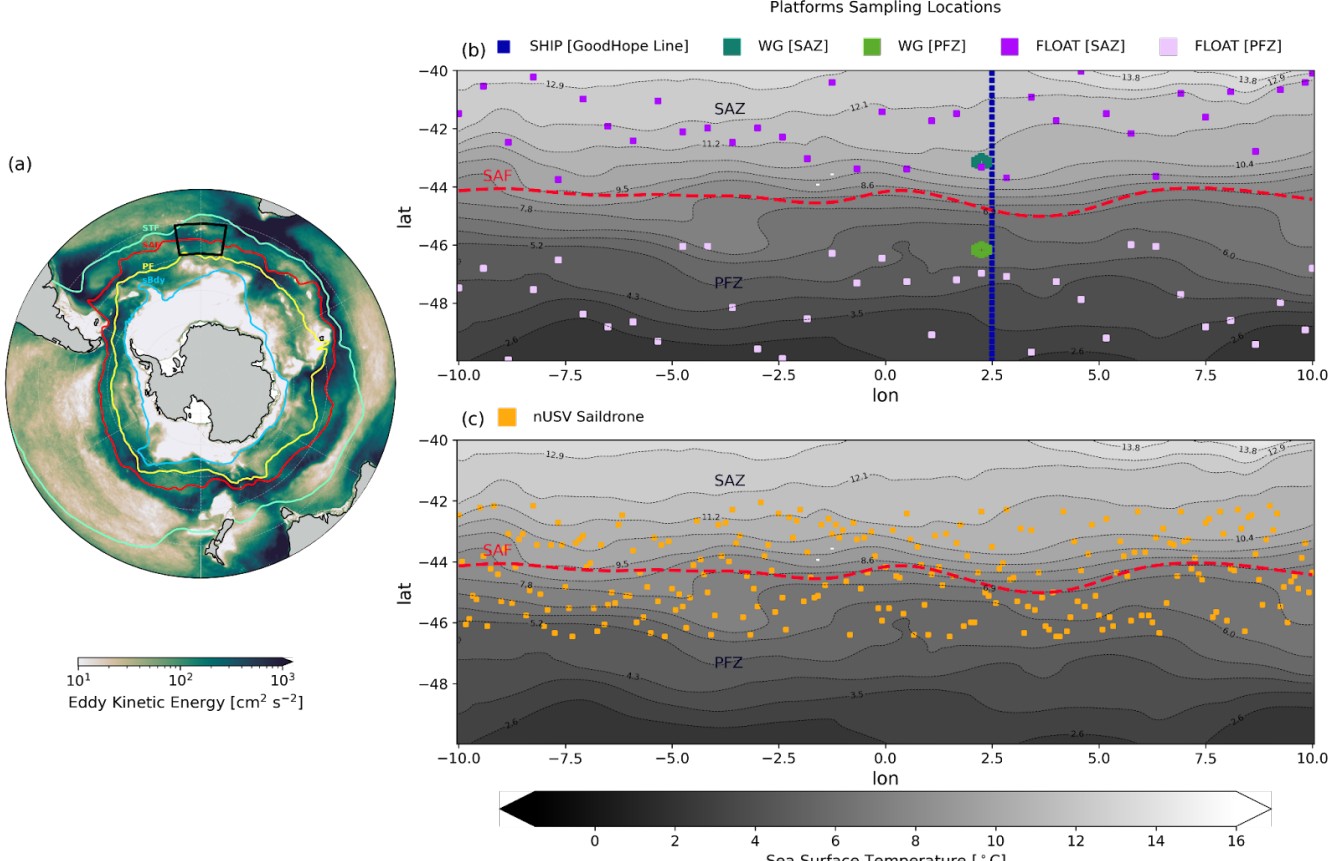

**Figure 1: Panel (a) is the regional view of the BIOPERIANT12 model simulations with the selected experimental domain (black box)**
**within the annual mean of the Southern Ocean major fronts and the changing conception of the Antarctic Circumpolar Current (ACC), showing the mean annual of eddy kinetic energy (EKE) derived from the model. From the north to south are the mean locations of the named fronts: the Subtropical Front (STF), the Subantarctic Front (SAF), the Polar Front (PF), and the Southern Boundary (SBdy) front (based on** Orsi et al., 1995)**. Colours show the EKE, illustrating the strong steering of the fronts. Panel (b) shows the map of the SST in the experimental domain (black box in Fig. 1a) on which are also shown the idealized sampling**

**tracks/locations of the synthetic ocean observing platforms, SHIP, FLOAT and WG as described in the figure legend. Panel (c) shows the sampling tracks of the idealized new unmanned surface vehicle (nUSV) Saildrone within the experimental domain. These locations, marked and coloured according to each corresponding sampling platform, are where we sample the BP12 model data in a way that is comparable to the real world. The SAF is characterized by the red line (Fig. 1a) and red dashed line (Fig. 1b-c), and it separates the experimental domain into the Sub-Antarctic Zone (SAZ) and Polar Front Zone (PFZ).**

This study was designed as a semi-idealized observing system simulation experiment (OSSE) to minimize some of the potential confounding factors on the final estimation of the root mean square error (RMSE), mean absolute (MAE) and temporal and spatial biases while evaluating the performance of regression models used to extrapolate surface ocean $pCO_2$ values. A key part of this design was to remove the normal step of clustering whose at large-scale mapping domain is necessary to overcome

the spatial and temporal limitations of observations (Fay and McKinley, 2013, 2014; Gregor et al., 2019; Landschützer et al., 2014). Thus, to avoid the clustering step, we chose a domain in the high-resolution (±10km) BP12 forced ocean biogeochemical model that was not only spatially and temporally coherent but also big enough to reflect the spatial and temporal scales necessary to provide sufficient sensitivity to the different sampling strategies. The selected domain, 10 degrees of latitude (40ºS - 50ºS) and 20 degrees of longitude (10ºW - 10ºE) as depicted in Fig. 1a, is in the Atlantic sector of the Antarctic

Circumpolar Current (ACC) between the Subtropical Front (STF) and the Polar Front (PF) and spans across the Sub-Antarctic Front (SAF) (Fig. 1a). Furthermore, the domain lies within the Sub-Polar Seasonally Stratified (SPSS) biome (Fay and McKinley, 2014). The Good Hope repeat hydrography sampling line passes through the domain (Fig. S1) for which sustained annual to bi-annual ship-based observations have been carried out for over a decade, as well as high-resolution carbon glider observations (Monteiro et al., 2015). More specifically, as shown in Fig. 1, our selected domain is crossed by the SAF,

therefore, includes the SAZ and the PFZ, inspired by (Gray et al., 2018) and (Chapman et al., 2020). The SAZ and PFZ, separated by the SAF (red line Fig. 1a, and red dashed curve in Fig. 1b-c), are respectively referred to as the north and south of the experimental domain.

The oceanographic context of this domain is shown in Fig. 1a, depicting the selected 10º-by-20º domain (black box) in the

200 context of the Southern Ocean major fronts and the eddy kinetic energy (EKE) derived from the BP12 model. This confirms that the domain spans the Sub-Antarctic Front (SAF) and is in a region of relatively high/medium EKE.

### 2.3.2 Model vs data products: the mean seasonal cycle of $pCO_2$

The mean seasonal cycles of $pCO_2$ reconstructions from two well-known machine-learning-based products (Landschutzer et

al., 2016; Gregor et al., 2019) are explored here within the study sub-domain in comparison with the BP12 model $pCO_2$ (Fig. 2). In the Southern Ocean, the observed maximum-positive anomaly in surface ocean $pCO_2$ in winter (Jul - Sep) is linked to mixed layer deepening and associated entrainment, while the maximum-negative anomaly in summer is linked to the spring-summer net primary production (Gregor et al., 2018; Takahashi et al., 2009).

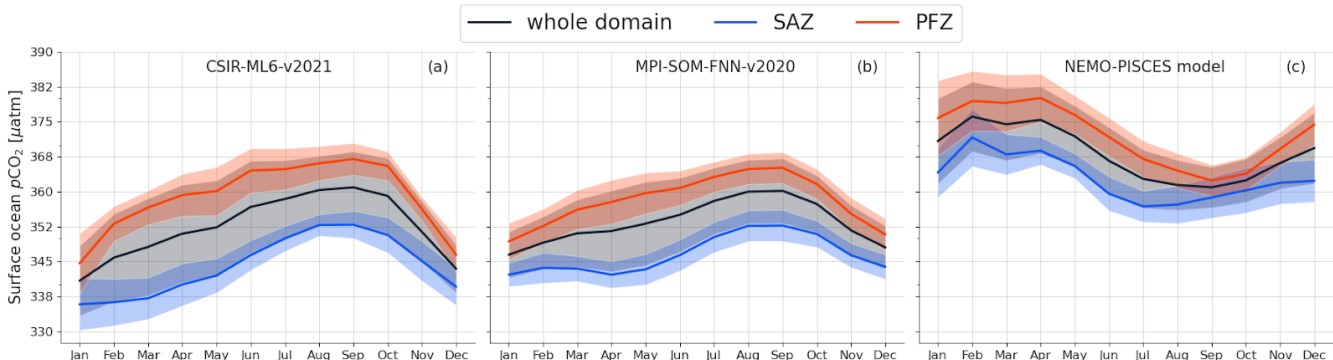

**Figure 2: The mean seasonal cycle (SC) for surface ocean $pCO_2$ from two observation-based products and a high resolution (±10km) forced NEMO-PISCES ocean model (BIOPERIANT12) within the selected experimental domain (Fig. 1a). It contrasts their respective seasonal cycles. Panels (a) and (b) respectively show the mean SCs of the $pCO_2$ estimates from the two data products: CSIR-ML6-v2021** (Gregor et al., 2019) **and MPI-SOM-FNN-v2020** (Landschützer et al., 2016) **in the whole domain, the SAZ, and the PFZ; and similarly, panel (c) shows the mean SCs of the $pCO_2$ from the BIOPERIANT12 model.**

The BP12 model sub-domain (black box, Fig. 1a) is depicted as a winter-maximum and summer-minimum $pCO_2$ area by both data products (Gray et al., 2018; Keppler and Landschützer, 2019) as shown in Fig. 2a-b. Thus, the domain-mean seasonal cycles of $pCO_2$ from these two products are quite consistent with the broader Southern Ocean (Gregor and Gruber, 2021). This is in sharp contrast with the seasonal cycle climatology from the high-resolution forced ocean model used in this study (Fig. 2c). The basis for this difference is that the high-resolution forced ocean model has a seasonal cycle that is largely influenced by the annual cycle of SST (Fig. 2c). This kind of temperature-driven model bias for surface ocean $pCO_2$ is now well recognized in both forced and coupled models in the Southern Ocean (Mongwe et al., 2016, 2018), but this study is more concerned with the modes of variability than it is with the mechanisms within the model. The forced coupled ocean model (NEMO-PISCES) represents the processes that regulate CO2. However, for the purpose of this study, the "correctness" of the $pCO_2$ response to the driver variables does not really matter because here we examine the sensitivity of the reconstruction to how the sampling scales match the modes of variability.

### 2.3.3 Synthetic ocean observing platforms

In designing the sampling scales and strategies we opted to constrain the experiment to realistic and existing observing platforms that can make direct $pCO_2^{ocean}$ or derived (from pH) surface ocean CO2 observations. More specifically, the existing ocean observing platforms involved in these experiments are the ships (serving as the baseline), and the following autonomous unmanned surface vehicles (USVs): Carbon-floats, Wavegliders and Saildrones (the new USV), whose simulations are dubbed SHIP, FLOAT, WG, and nUSV, respectively (Fig. 1b-c). The first autonomous platform, Carbon-float, characterizes the

autonomous profiling biogeochemical float operating in the Southern Ocean (Majkut et al., 2014; Williams et al., 2017; Gray et al., 2018). Manufactured by Teledyne/Webb Research or Seabird Electronics, these floats are designed to provided year-round measurements at 10-day periods (Johnson et al., 2017). The second autonomous platform, Waveglider, is an autonomous USV developed by Liquid Robotics Inc (Sunnyvale, California, USA), that is unique in its ability to harness ocean wave and solar energy for platform propulsion (Hine et al., 2009). At sea it operates individually or in fleets delivering real-time data for several months without servicing (Grare et al., 2021; Sabine et al., 2020). Equipped with physical and biogeochemical instruments/sensors, the Waveglider gathers thus ocean data in ways or locations previously either too costly or challenging to operate. Made by Saildrone Inc (Alameda, California, USA), the nUSV Saildrone is an autonomous ocean-going data collection platform navigable via satellite communications and designed for long-range, long duration missions of up to 12 months (Gentemann et al., 2020; Meinig et al., 2016, 2019). It is predominantly powered by wind and solar energy, and equipped with meteorological, ocean physical and biogeochemical sensors for long-range ocean data collection missions (Gentemann et al., 2020) through remote surveying in the toughest of ocean environments such as the Southern Ocean (Meinig et al., 2019; Sutton et al., 2021).

Each of these simulated ocean observing platforms had a sampling routing through the domain that closely approximated reality. Ship-based sampling is along a single meridional repeat line (longitude), where repeats could be seasonal and annual (Fig. 1b). Floats followed a zonal sampling distribution that is consistent with the flow of the ACC and a 10-day sampling scale with a limited random meridional mesoscale variability which reflects the eddy kinetic energy (EKE) characteristics of the domain but is constrained by the SAF (Fig. 1a-b). Wavegliders were constrained to repeat the pseudo-mooring sampling (±20km range) on the ship line (Fig. 1b), which captures the sub-mesoscale gradients but with a high temporal sampling frequency of 1 hour. Moreover, from a logistic perspective, WGs were given a mooring-like sampling program to ease their deployment and retrieval, for example, from the research vessel SA Agulhas II which crosses the domain at the Good Hope line, whereas nUSVs would be able to sail to the next port.

### 2.3.4 Idealized experiment setup

In this paragraph, we briefly describe the experimental scenarios shown in Table S2. We stress again on the fact that these experiments are intentionally made to reproduce the sampling resolutions of their real-world counterparts, not necessarily the spatial resolution in practice but at least the temporal one. We considered the NEMO-PISCES model simulations, BP12, as a realistic representation of the real ocean climate systems within which the $p\text{CO}_2^{\text{ocean}}$ is known across the entire experimental domain. Based on this, we ask the following question: given measurements of $p\text{CO}_2^{\text{ocean}}$ as sampled in a real-world scenario by these ocean observing platforms, how sensitive is the sampling distribution and resolution to observation-based estimates of $p\text{CO}_2^{\text{ocean}}$ at every point across the entire experimental domain?

In these experiments, we simulate the sampling tracks/patterns of the synthetic ocean observing platforms SHIP, FLOAT, WG, and nUSV Saildrone (Fig. 1b-c). We leverage these synthetic sampling systems to sample the BP12 model data inside our selected experimental domain by constraining the experiment to their realistic and existing counterparts. The BP12 model sampled data from each of the sampling scenarios are then used for training and testing of the ML algorithms. The trained ML models are used to reconstruct the $pCO_2^{ocean}$ values of the full experimental domain and compared with original BP12 model field $pCO_2^{ocean}$ to assess the anomalies in reconstructed mean annual and seasonal cycles.

The idealized ship operates according to the sampling scales and strategies of ships involved in SOCAT collaborative effort. However, here we considered the three following seasonal sampling regimes for the ship platform: (1) summer only, (2) winter and summer, and (3) autumn and spring. Like the real-world scenario, the ship simulation served as our baseline. The idealized carbon-float simulates SOCCOM biogeochemical float with a 10-day sampling cycle. (Talley et al., 2019) reported the importance of the water masses and frontal structures in the deployment strategy of autonomous sampling platforms, such as floats, that will likely follow the fronts with an eastward trajectory but will seldom cross the front. Therefore, we consider the situation where the idealized floats do not cross the SAF as illustrated in Fig. 1b, even though in reality this might happen due to the occurrence of events such as storms or eddies. We thus considered two deployment and sampling scenarios to not disadvantage the floats and to value their large spatial structure: (1) in the SAZ, and (2) in the PFZ (Fig. 1b). Given the pseudo-Lagrangian sampling patterns of an Argo float whose motion is driven by water current, we assume that our idealized float moves eastward and on a trajectory that is a Brownian motion or, more specifically, a random walk (Fig. 1b). The idealized Waveglider operates according to the sampling strategies of the Wavegliders used in the SOSCEx project (cf. Sect. S1.1 for additional details). Like the idealized float, we considered two deployment stations, the first in the SAZ (cf. Fig. 1b, hexagonal patterns in dark green) and the second in the PFZ (cf. Fig. 1b, hexagonal patterns in light green). This idealizes the two deployment scenarios of SOSCEx III gliders (cf. Fig. S1, hexagonal patterns in blue-yellow) that sampled on an hourly basis. However, given the model temporal resolution that is daily, our idealized Wavegliders samples daily. Lastly, we add an idealized Saildrone that simulates the sampling strategies of its real-world counterpart that can sample for up to 12-month ocean data collection missions (Gentemann et al., 2020; Meinig et al., 2019). As with the idealized Waveglider, the Saildrone also samples daily. Further, we assume that by leveraging its speed the Saildrone sampling can be done across an ocean front, such as the SAF as depicted in Fig. 1c – a realistic assumption because in reality nUSV Saildrones sample at a much higher frequency (hourly) and can be piloted remotely (Gentemann et al., 2020; Sutton et al., 2021). We assumed that all three autonomous sampling platforms sampled year-round in our experimental domain.

The observing system simulation experiment (OSSE) with nUSV Saildrone is inspired by the study of Sutton et al. (2021) that used nUSV to sample at a very high resolution and completed in about 6 months the first autonomous circumnavigation of Antarctica providing hourly observations. At this frequency, the nUSV sampling density in this study domain (Fig. 1c) is realistic due to the size of the sampling domain. We extracted a subset of Sutton et al. (2021)-USV dataset within the subdomain

to get its tracks (cf. Fig. S6) and found that the Sutton et al. (2021)-USV would take ~16 days to cover our 20ºW-E domain, which corresponds to 16days * 24hrs = 384 hourly samples. However, our nUSV sampling pattern (Fig. 1c) is idealized, with the goal of sampling across the sub-domains on both sides of the front (SAF); that is, in the SAZ and PFZ. By using a back-of-the-envelope approach, we find that the Saildrone would be able to cover our domain in 45 days using a zig-zag pattern - assuming 42°S to 46°S with each pass covering 2.5°W-E for each pass (~500 km) with 8 passes in our domain (4000 km) at a speed of ~2 knots (~3.7 km/hr).

In summary, we sample the $p\text{CO}_2^{\text{ocean}}$ and drivers using these above-mentioned synthetic sampling platforms, i.e., SHIP, FLOAT, WG, and nUSV Saildrone (Fig. 1b-c). We emphasize that these experiments are intentionally made to reproduce the sampling resolution of their real-world counterparts, not necessarily their spatial resolution in practice but at least the temporal one. Then we use ML regression techniques to reconstruct the full experimental domain and compare it with the BP12 model truth $p\text{CO}_2^{\text{ocean}}$ in the full domain to assess the reconstructions as anomalies of mean annual and seasonal cycles, which is a key objective of this work.

## 2.4 Machine Learning implementation

We use a two-member ensemble method (we call ML2) that consists of two state-of-the-art ML approaches: the Feed-forward Neural Network (FNN) and a variant of Gradient Boosting Decision Tree (GBDT) learning frameworks called Gradient Boosting Machines (GBM). Our choice of FNN method is motivated by its recent success in approximating the surface ocean $p\text{CO}_2$ (Denvil-Sommer et al., 2019; Gregor et al., 2019; Landschützer et al., 2013, 2016). The choice of the GBDT approach is motivated by its achievement of state-of-the-art performances in many ML tasks (Ke et al., 2017), and also the success of GBDT's previous approaches (Gloege et al., 2021; Gregor et al., 2019; Gregor and Gruber, 2021). We use the Scikit-learn and LightGBM Python packages for our implementation of FNN and GBM, respectively. We thus focus here only on the ensemble average ML2 whose stacking process is illustrated in Fig. 3a. Unlike the two main techniques of reference (Landschutzer et al, 2016; Gregor et al., 2019) both of which include a clustering step, in this study we avoided it because of the size of the study domain (10 degrees of latitude, 40ºS – 50ºS, by 20 degrees of longitude, 10ºW – 10ºE). More details of our motive of skipping this step is provided in S2.3.

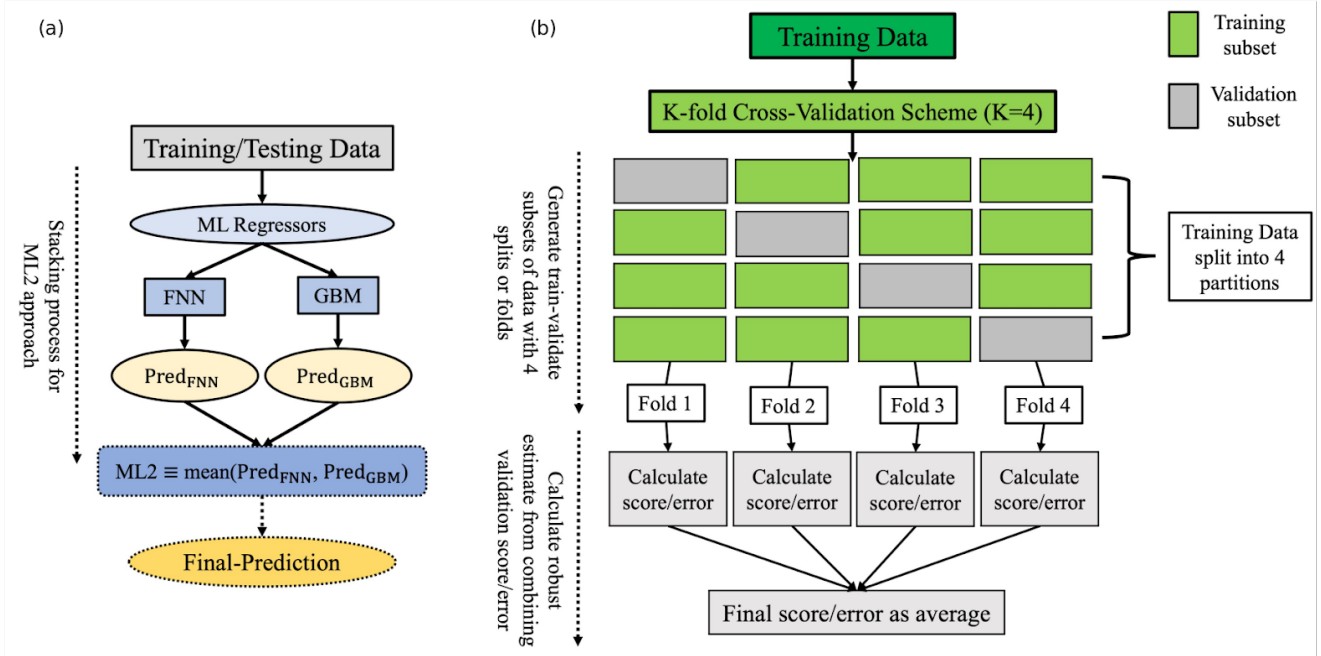

**Figure 3: Schematic flow diagram of the two-member ensemble method ML2. Panel (a) shows the schematic representation of the stacking process of the two machine learning (ML) algorithms, FNN and GBM, that make up ML2; panel (b) shows the schematic flow diagram of the K-fold cross-validation (CV) procedure used in hyper-parameter optimization (HOP) of the two members (FNN, GBM) of ML2. To extrapolate from surface ocean $pCO_2$ samples, ML2 uses full domain coverage model data of the predictor variables: SST, SSS, MLD) and Chl-a. These variables serve as proxies for known processes that affect surface ocean $pCO_2$** (Takahashi et al., 1993)**.**

Given that the observation size is relatively small, especially for the baseline experiment (SHIP summer-only), immediately split the simulated data into training and testing sets may not capture some key features of the original platform observations. We thus use the entire sampled data for model building instead of splitting the data into two sets. As shown in Fig. 3b, however, to control the overfitting, we incorporate a K-fold cross-validation (CV, with K=4) during the model training in order to find the set of hyper-parameters that enable a better generalization of ML2. Like in (Gregor et al., 2019), the CV is applied identically to each of the two-member algorithms (FNN and GBM) except that here, the tuning of hyper-parameters was achieved using a Bayes-search CV (BayesSearchCV) instead of a grid-search CV. We make use of the Scikit-optimize Python package for our BayesSearchCV implementation. The optimal values of ML2 hyper-parameters used were reported at the end of the training and included in the Supplementary Materials (Tables S4 and S5) for reproducibility. Further, the testing of generalization is done through quantitative comparison of the estimates with model data (known truth) that were not involved in the simulations of synthetic platforms.

## 2.5 Machine learning regression metrics

Although the choice of the performance measure may seem straightforward and objective, it is often difficult to choose a metric that corresponds well to the desired behaviour of the ML algorithm (Goodfellow et al., 2016). The reconstruction power of the surface ocean $pCO_2$ of the full experimental domain are thus estimated using a series of four statistical metrics that include the mean bias error (MBE), mean absolute error (MAE), root mean square error (RMSE), and Pearson's correlation coefficient ($r$) to measure the tendency or strength of estimates and observations to vary together (Stow et al., 2009) or, more

technically, to quantify the level at which reconstruction captures the phasing observed in the model truth (Gloege et al., 2021).

The MBE, commonly called bias, is the mean difference between the estimates and the target variable samples. It captures the average bias/error in the predictions and is calculated as follows:

$$MBE = \frac{1}{n}\sum_{i=1}^{n}(\hat{y}_i - y_i), \tag{2}$$

where $n$ is the number of samples, $\hat{y}$ is the model prediction and $y$ is the target variable (in this case, $pCO_2^{ocean}$).

The MAE denotes the ratio of the L1 norm of the error vector to the number of samples ($n$). More specifically, the MAE derives from the unaltered magnitude (or absolute value) and provides an estimate of the average magnitude of the error. It is calculated as follows:

$$MAE = \frac{1}{n}\sum_{i=1}^{n}|\hat{y}_i - y_i| \tag{3}$$

The RMSE, one of the most popularly used metrics in the climatic and environmental sciences community when dealing with regression modelling problems, is also a measure of the difference between the estimates $\hat{y}_i$ and the target variable samples $y_i$. It provides an estimate of the variability in the predictions in terms of the fitness with the observed data, and is defined as

follows:

$$RMSE = \sqrt{\frac{1}{n}\sum_{i=1}^{n}(\hat{y}_i - y_i)^2} = \sqrt{MSE} \tag{4}$$

Where MSE is simply the mean square error. For squaring individual errors $e_i = \hat{y}_i - y_i$ ($i = 1,\ldots,n$), the stated rationale is usually to "disconnect the sign" of $e_i$ so that the magnitudes of errors influence the average error, MSE.

In order to quantify the strength of the linear association between the $pCO_2^{ocean}$ estimates (i.e., $\hat{y}_i$ for $i,\ldots,n$) and observations/known truth (i.e., $y_i$ for $i = 1,\ldots,n$), the Pearson's correlation coefficient ($r$) is used. Its computing is formulated as follows:

$$r = \frac{1}{(n-1)\sigma_y\sigma_{\hat{y}}}\sum_{i=1}^{n}(y_i - \bar{y})(\hat{y}_i - \bar{\hat{y}}), \tag{5}$$

where $\sigma_y$ and $\sigma_{\hat{y}}$ are the standard deviations of $y$ and $\hat{y}$, respectively; $\bar{y}$ and $\bar{\hat{y}}$ the means of $y$ and $\hat{y}$, respectively. The correlation coefficient always takes values between -1 and 1, with lower (near -1) and higher (near 1) values of $r$ respectively indicative of how much reconstruction and model are in or out of phase. Values of $r$ that are close to 0 are indicative of no association between the two signals. Therefore, the ideal value for $r$ will be close to one.

## 2.6 Uncertainty decomposition/breakdown

A firm understanding of the uncertainties is required for the purpose of our analysis given that in our study we are dealing with the uncertainties that we cannot fully quantify now as this is on unseen or out-of-sample data like in (Gloege et al., 2021). Therefore, it is necessary to distinguish the different types of uncertainties. We assume that our sampled observations are unbiased, and hence the training data sets for surface ocean $pCO_2$ are considered such as known; and this can be justified by the fact that we have access to the whole data. The terms error and uncertainty are interchangeably used although here the latter is used as an estimate quantifiable against a known value whereas the former characterizes a range of values within which the true value is asserted to lie with some level of confidence (Gregor and Gruber, 2021).

The $pCO_2$ total uncertainty (E) is dealt with as in (Gregor and Gruber, 2021). The authors identified three main sources of errors that contribute to E within the surface ocean carbonate system. This includes (1) the measurement (M), (2) representation (R), and (3) prediction (P) errors. Under the assumption that these components are independent of each other in the $pCO_2$ total uncertainty space, E can thus equivalently be expressed as the norm of the vector whose coordinates are P, M and R; that is, the square root of the sum of the squares of these components: $E = \sqrt{P^2 + M^2 + R^2}$. We can remove the contribution of the measurement uncertainty from this equation since we are sampling from a synthetic dataset. Further, we address the representation uncertainty by sampling at a higher resolution (Gregor & Gruber, 2021). Given that we are predicting at high resolution (1/12° daily), the sampling distribution bias due to capturing of large-scale gradients is assumed to be small since we are within the 2-day threshold set by (Monteiro et al., 2015). Lastly, we assume that the ML models are the best possible predictors for the given training datasets, since each ML model was trained using best practices (i.e., low in-sample errors calculated from all the training points as shown in the Supplementary Materials). Therefore, reported RMSEs will be the uncertainties due to sampling bias.

## 3 Results

In the next sections, the results for the following four sets of semi-idealized model experiments combinations, SHIP, SHIP + FLOAT, SHIP + WG, and SHIP + nUSV are presented in terms of spatial and seasonal cycle anomalies of the annual mean $pCO_2$ estimates.

### 3.1 Annual mean seasonal cycle for the domain

The annual mean map for $pCO_2$ (mean 368.15 µatm; standard deviation 50.5 µatm) shows that the domain is characterized by both meridional and mesoscale variability expected from the mesoscale resolving BIOPERIANT12 model (Fig. 4a). The meridionally distinct SAZ (north of the domain) (< 368.15 µatm) and PFZ (south of the domain) (> 368 µatm), are separated by the Sub-Antarctic Front (SAF) (Figs. 1 and 4a). This mean map also highlights the importance of mesoscale gradients in both the SAZ and the PFZ domains (Fig. 4a). The mean seasonal cycles of $pCO_2$ for the whole domain as well as for the SAZ (lower - blue) and PFZ (higher - red) are depicted in Fig. 4b. It shows that the seasonal cycle of $pCO_2^{ocean}$ is dominated by the influence of the annual cycle of the sea surface temperature (SST) on $CO_2$ solubility (Mongwe et al., 2016; Munro et al., 2015) with warm late summer (Feb-Apr) and cool late winter (Jul-Sep) (Fig. 4b). The three seasonal cycles (whole domain, SAZ, and PFZ) show coherence in the seasonal amplitude and phasing except that the warming transition from winter to spring occurs two months earlier (Jul) in the SAZ relative to the PFZ (Fig. 4b).

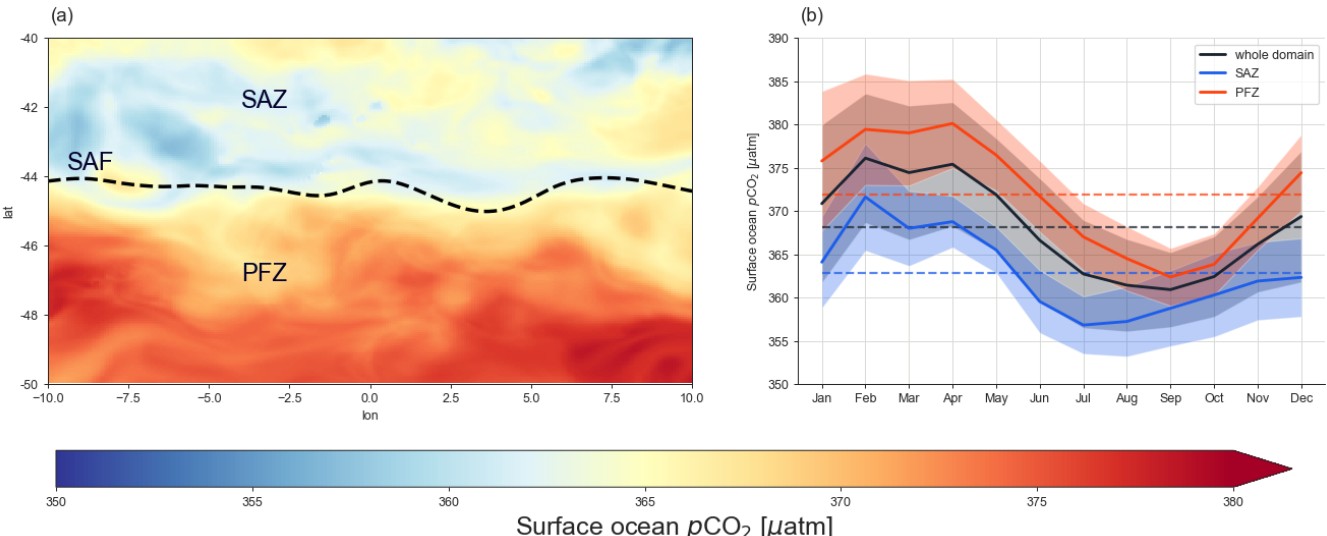

**Figure 4: Characterisation of the spatial and temporal surface ocean $pCO_2$ annual mean state within the selected 10º-by-20º experimental domain located in the northern ACC that corresponds to SPSS biome (Fay and McKinley, 2014) as shown in Fig. 1a. Panel (a) shows the map of mean annual $pCO_2$ from the BIOPERIANT12 (BP12) model. It shows that the domain is characterized by a regional meridional gradient including the Sub-Antarctic Front (SAF) (black dashed line) as well as mesoscale gradients in both SAZ and PFZ; panel (b) shows the mean seasonal cycles for surface ocean $pCO_2$ in the BP12 model domains (SAZ, SAF and PFZ) where the dashed lines indicate the magnitude of the annual mean –or each domain - 368.16 µatm (domain), 362.85 µatm (SAZ), and 371.78 µatm (PFZ).**

Notwithstanding the phasing differences, we still find a comparable winter reconstruction bias in this study (Figs. 2c and 4b) and observation-based products (Fig. 2a-b). Thus, the question is: is the magnitude of the reconstructed winter $pCO_2$ maximum

realistic or a result of the way the machine learning methods process the summer sampling bias in a system characterized by strong seasonal and intra-seasonal modes of variability?

## 3.2 Reconstructed mean annual spatial and seasonal cycle anomalies

In order to investigate the anomalies in the reconstruction of the mean annual and seasonal cycles, which are a key objective of this study, we first characterized the anomaly by the mean bias error (MBE) and calculated the MBE at each grid point of the spatial domain. Secondly, we also calculated the anomaly of the seasonal cycle reconstruction in each of the sub-domains. More specifically, we used the seasonal cycle residuals to explore how a systematic anomaly could influence the reconstruction of $p\mathrm{CO}_2$ values at the surface ocean. We performed this calculation for each experiment and their respective reconstructions and also examined their spatial variability.

### 3.2.1 Semi-idealized SHIP-only observations experiment results

The semi-idealized SHIP-only sampling experiments mimic the largely ship-based SOCAT gridded product to evaluate the sensitivity of the reconstruction uncertainties (RMSE, MAE, MBE/bias) to seasonal meridional sampling scenarios. In each of these scenarios the ship makes two meridional crossings in opposite directions one month apart (Fig. 1b). This SHIP-only set of seasonal sampling experiments is our baseline as it is also used in all platform combinations. Three seasonal sampling scenarios (summer (smr), summer+winter (smr+wtr), and autumn+spring (aut+spr)) were considered. While the first two scenarios are addressed in detail in this study (Fig. 5a-b, and Table 1), the third one can be found in the Supplementary Material (Fig. S5, and Table S6) in support of the main points already made in Fig. 5a-b.

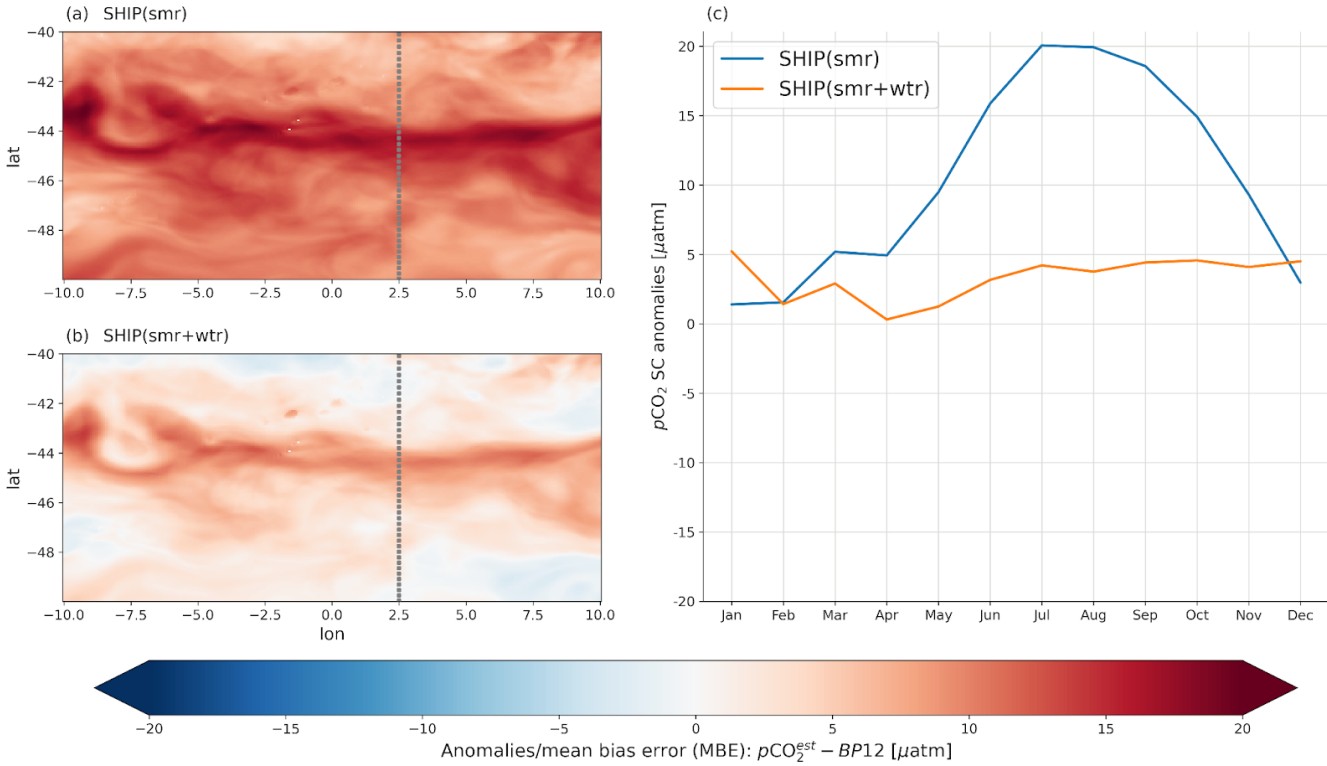

**Figure 5: Reconstruction anomalies for the idealized SHIP experiment where the idealized ship sampled the domain based on the sampling regimes/scenarios, SHIP(smr) for summer, SHIP(smr+wtr) for summer and winter. Panels (a) and (b) show the maps of the reconstruction anomalies according to the two sampling regimes SHIP(smr), and SHIP(smr+wtr) respectively; panel (c) shows the anomalies of the mean seasonal cycle (SC) reconstruction based on these two sampling regimes; that is, SHIP(smr) and SHIP(smr+wtr). The meridional dotted grey line in panels (a) and (b) illustrates the sampling line (summer & winter) and serves as a reminder of how SHIP sampling was performed.**

The spatial and seasonal cycle anomalies from the reconstructions for the summer (smr), summer and winter (smr+wtr) sampling lines are depicted in Fig. 5a-b. The results for the autumn and spring (aut+spr) sampling lines are summarized in the Supplementary Materials (Fig. S6). The uncertainties and regression errors for all three experiments are shown in Table 1. These results showed that the highest positive anomalies in the reconstruction of the mean and the seasonal cycle occurs when a ship, samples (i.e., makes 2 passes in consecutive months) the sub-domain only in summer (Fig. 5a, c). This sampling strategy resulted in a strong positive anomaly (±20 µatm) that peaks in winter and weakens in mid-summer (Fig. 5c). In sharp contrast, when winter sampling crossings are added to the summer scenario (smr+wtr) the spatial and seasonal anomalies are significantly reduced from 20 µatm to < 5 µatm respectively (Fig. 5b, c). The weaker but persistent positive anomaly in the SAF accounts for most of the reduced positive seasonal cycle anomaly (Fig. 5a, c).

All scenarios depict a mesoscale modulated positive annual $pCO_2$ anomaly (MBE) climatology in the vicinity of the SAF (Fig. 5a-b). However, this is slightly offset by equally strong positive anomalies in the SAZ and PFZ for the smr scenario (Fig. 5a),

while the meridional gradients of the anomalies are much weaker for the smr+wtr scenario (Fig. 5b). These differences are very well reflected in the anomalies of their corresponding seasonal cycles (Fig. 5c).

**Table 1: ML regression modelling scores of the ensemble average (ML2) for two sampling scenarios of SHIP experiment: SHIP(smr)**
**for summer sampling, and SHIP(smr+wtr) for summer and winter sampling. The configuration of this set of experiments is presented in Table S2 and clearly described in Sect. 2.3.4. The first column of the table is the experimental set and the second one corresponds to the considered experiments. The statistical metrics used to assess ML2 for this set of experiments are abbreviated as follows: RMSE is the root mean square error calculated following Eq. (4); MAE is the mean absolute error (Eq. 3); MBE or Bias is the mean average error (Eq. 2); and $r$ is the Pearson's correlation coefficient (Eq. 5) between the reconstructed and BP12 model truth $pCO_2$. Values in the table are significantly different from the mean for the corresponding column (with a 95% confidence level**
**or p-value < 0.05 for the two-tailed Z-test).**

| Set | Experiments | RMSE (µatm) | MAE (µatm) | MBE (µatm) | $r$ |
|---|---|---|---|---|---|
| SHIP | SHIP(smr) | 13.79 | 11.51 | 10.52 | 0.36 |
| | SHIP(smr+wtr) | 6.8 | 5.29 | 3.18 | 0.73 |

These SHIP-only experiment results (Tables 1 and S3) also show that the summer-only sampling of the sub-domain both produces the largest sampling bias (10.52 µatm, with an RMSE of 13.79 µatm) and yields the weakest correlation between the underlying $pCO_2$ estimates and the model ground-truth (with $r = 0.36$). On the other hand, it also showed that if the ship
undertakes just one more meridional voyage in winter, this halved the RMSE to 6.8 µatm and the bias (MBE) to 3.18 µatm compared to the summer-only sampling experiment, SHIP(smr). Moreover, it also strengthened the linear association between the reconstruction and BP12 model ground-truth for $pCO_2$ ($r = 0.73$).

### 3.2.2 Idealized SHIP and autonomous observations platform experiments

In this section, we presented the results of three sets of combined ship and autonomous platform experiments (SHIP(smr) + FLOAT, SHIP(smr) + WG, and SHIP(smr) + nUSV) that allowed us to test the hypothesis that complementing summer biased ship-based sampling with year-long high-resolution sampling in space and time reduces the reconstruction uncertainties and positive annual mean and seasonal cycle biases relative to the ship-sampling alone (Figs. 5a, c, and 6a-b)(Bushinsky et al., 2019; Gregor et al., 2019; Sutton et al., 2021). We simulated and analysed the reconstruction of the mean annual $pCO_2$ and
seasonal cycles from carbon-floats (FLOAT) and carbon Wavegliders (WG) deployed independently for a year in the Sub-Antarctic Zone (SAZ) and Polar Frontal Zone (PFZ) (Figs. 5b, 5c-d, and 5e-f). These were complemented by simultaneous year-round FLOAT deployments in the SAZ and PFZ (Fig. 5b, g), and a deployment of the new unmanned surface vehicle (nUSV) Saildrone that spanned across all three domains (Fig. 5b, h).

These results show that both the reconstructed mean annual anomaly and the seasonal cycle of $pCO_2$ are very sensitive to the spatial and temporal characteristics of the additional autonomous sampling platform (Fig. 6). Statistics (Table 2) show that all the autonomous platform deployments experiments improved the significant winter positive biased seasonal cycle anomaly from the summer ship sampling reconstruction (±20 µatm). However there remained a small but variable (2 - 10 µatm) winter - spring seasonal bias in all deployment combinations (Fig. 6b). The exception was the experiment with a FLOAT deployment

in the SAZ, which resulted in a negative seasonal bias that also peaked in winter (±10 µatm) and started earlier in the autumn (Fig. 6b). The two experiments with the smallest seasonal biases were the SHIP(smr) + WG (SAZ), and SHIP(smr) + nUSV. The first, SHIP(smr) + WG(SAZ), showed a small negative bias in the summer (< -5 µatm) and a small positive bias in the winter (< 5 µatm). The latter, SHIP(smr) + nUSV, showed a small positive bias in summer (0 - 5 µatm) and in winter (4 - 5atm)(Fig. 6b). In contrast, the experiment SHIP(smr) + FLOAT(SAZ+PFZ) that combined the two year-round FLOAT

deployments (SAZ and PFZ ), shows a minimal bias in summer but among the highest for all the experiments in winter (±10 µatm)(Fig. 6b).

The spatial annual mean $pCO_2$ experimental scenario anomalies are consistent with the characteristics of the seasonal cycle of $pCO_2$ (Fig. 6a, 6c-h). In all cases the Sub-Antarctic Front (SAF) emerged as a feature with a variable positive $pCO_2$ anomaly

relative to the SAZ and PFZ sectors to the north and south respectively (Fig. 6a, 6c-h). All the scenarios highlight significant mesoscale anomaly gradients across all the domains (Fig. 6a, 6c-h). The year-long deployment of FLOATs and WGs in the SAZ lead to negative anomalies in both the SAZ and the PFZ but those for the WG experiments are significantly weaker (Fig. 6c, e).

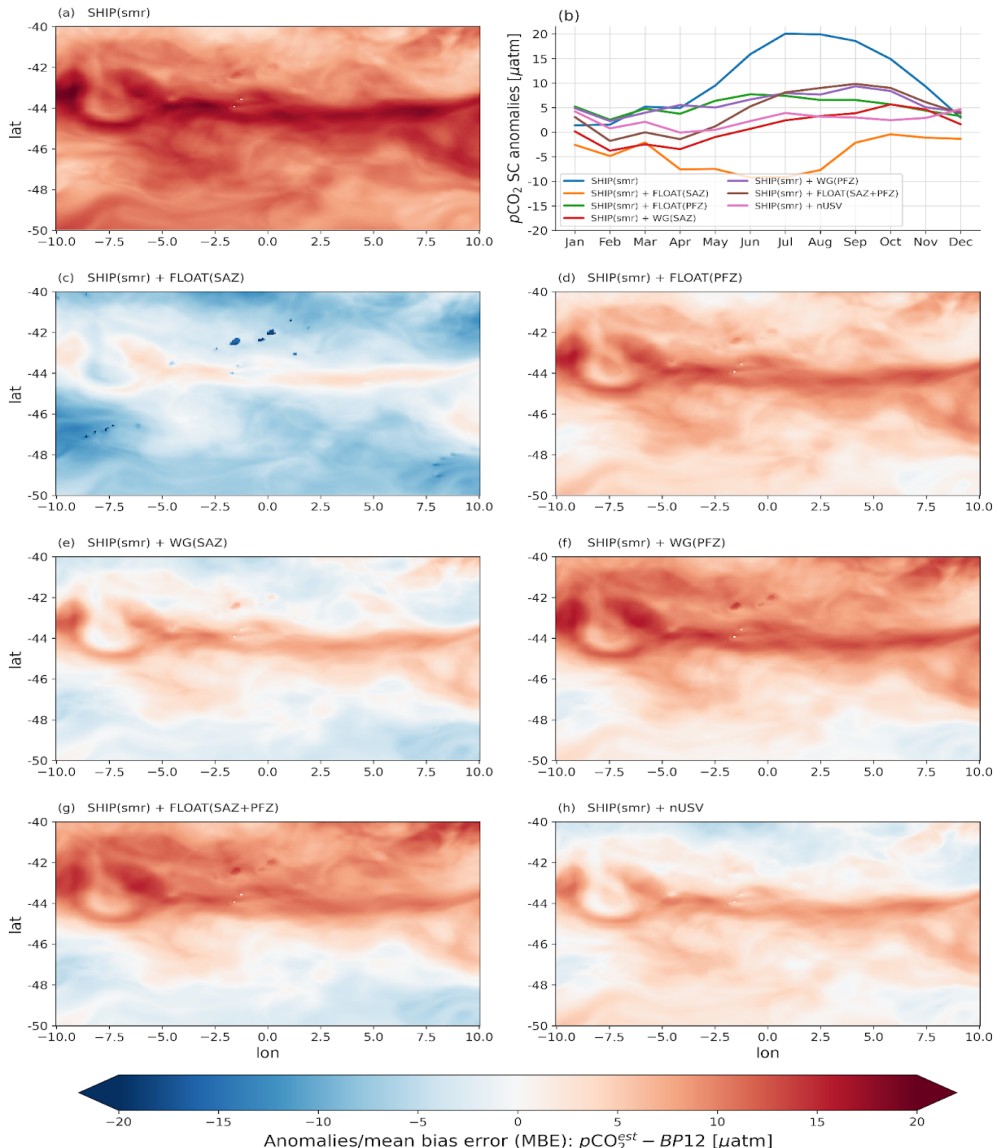

**Figure 6: Reconstruction anomalies for the 4 sets of experiments, SHIP, SHIP + FLOAT, SHIP + WG, and SHIP + nUSV with a particular focus on the summer-only baseline scenario: SHIP(smr). Panels (a) shows the spatial anomalies or biases (MBEs) of the mean annual $p$CO$_2$ reconstruction for the SHIP summer-only sampling scenario, that is, SHIP(smr); panel (b) shows the anomalies of mean seasonal cycle (SC) reconstructions of the summer-only sampling scenario of the above-mentioned sets of experiments; panels (c) and (d) show the spatial reconstruction anomalies for the SHIP + FLOAT experiments where two independent FLOATs were deployed in the SAZ and PFZ (respectively), and used to supplement SHIP(smr); panel (e) and (f) show the spatial reconstruction anomalies for the SHIP + WG experiments where two independent WGs were deployed along the SHIP line in the SAZ and PFZ (respectively), and used to supplement SHIP(smr); panel (g) shows the spatial anomalies of the mean annual $p$CO$_2$ reconstruction for the SHIP + FLOAT experiment scenario where the two FLOAT deployments (SAZ and PFZ) were used to supplement the SHIP summer-only sampling scenario, hence SHIP(smr) + FLOAT(SAZ+PFZ); and panel (h) shows the spatial anomalies of the mean annual $p$CO$_2$ reconstruction for the SHIP + nUSV experiments where SHIP(smr) were supplemented by a year-round sampling of the nUSV Saildrone, hence SHIP(smr) + nUSV.**

However, the reverse was found for the SAF zone which shows a stronger positive anomaly for the WG(SAZ) than for the FLOAT(SAZ) (Fig. 6c, e). The stronger mean annual negative $pCO_2$ anomaly for the SHIP(smr) + FLOAT(SAZ) deployment is consistent with the negative seasonal cycle anomaly, which points to the mean annual anomaly being mainly influenced by the winter negative anomaly (Fig. 6b-c). Similarly, the much weaker negative anomalies in the SAZ and PFZ for the WG deployment are consistent with the weaker seasonal cycle ($< \pm 5$ µatm) of $pCO_2$ for the whole domain.

SHIP(smr) + FLOAT(PFZ) and SHIP(smr) + WG(PFZ) deployments result in weak to moderate positive anomalies in the northern half of the PFZ, the SAZ and the SAF and weak to zero anomalies in the southern PFZ, all of which are characterized by mesoscale gradients (Fig. 6d, f). Both scenarios show a comparable positive seasonal cycle anomaly although the phasing of the winter maximum is earlier Jun vs Sep for the SHIP(smr) + FLOAT(PFZ) (Fig. 6b). The mean annual $pCO_2$, from the combined SHIP(smr) + FLOAT(SAZ+PFZ) deployments, showed spatial characteristics similar to the SHIP(smr) + FLOAT(PFZ) but with intensified negative and positive anomalies in the PFZ and SAZ respectively (Fig. 6g). The moderately strong positive winter anomalies ($\pm 10$ µatm) in the seasonal cycle for this experiment indicate that the mean annual positive anomalies are also dominated by the winter anomalies (Fig. 6b). The mean annual $pCO_2$ anomaly for the SHIP(smr) + nUSV deployments is weakly negative ($< -5$ µatm) in the north SAZ and weakly positive ($< 5$ µatm) in the SAF and the PFZ (Fig. 6h). The overall weak mean annual $pCO_2$ anomaly is consistent with the weakest ($0 - 5$ µatm) seasonal cycle anomaly (Fig. 6b).

**Table 2: ML regression modelling scores of the ensemble average (ML2) for the summer-only sampling scenario (smr) of all the 4 sets of experiments: SHIP, SHIP + WG, SHIP + FLOAT, and SHIP + nUSV. The configuration of these experiments is presented in Table S2 and described in Sect. 2.3.4. Similar to Table 1, the first column of the table is the experimental set and the second one corresponds to the considered experiments. The statistical metrics used to assess ML2 for this set of experiments are abbreviated as follows: RMSE is the root mean square error calculated following Eq. (4); MAE is the mean absolute error (Eq. 3); MBE or Bias is the mean average error (Eq. 2); and $r$ is the Pearson's correlation coefficient (Eq. 5) between the reconstructed and the BP12 model truth $pCO_2$. Values in the table are significantly different from the mean for the corresponding column (with a 95% confidence level or p-value < 0.05 for the two-tailed Z-test).**

| Sets | Experiments | RMSE (µatm) | MAE (µatm) | MBE (µatm) | $r$ |
|---|---|---|---|---|---|
| SHIP | SHIP(smr) | 13.79 | 11.51 | 10.52 | 0.36 |
| SHIP + FLOAT | SHIP(smr) + FLOAT(SAZ) | 9.29 | 7.46 | -4.81 | 0.60 |
| | SHIP(smr) + FLOAT(PFZ) | 8 | 6.51 | 5.32 | 0.73 |
| | SHIP(smr) + FLOAT(SAZ+PFZ) | 9.12 | 7.57 | 4.14 | 0.63 |
| SHIP + WG | SHIP(smr) WG(SAZ) | 6.88 | 5.4 | 0.82 | 0.64 |
| | SHIP(smr) WG(PFZ) | 9.41 | 7.59 | 5.88 | 0.57 |
| SHIP + nUSV | SHIP(smr) + nUSV | 6.4 | 5.1 | 2.38 | 0.74 |

Table 2 shows that SHIP(smr), the baseline biased ship-summer sampling experiment (the status quo in the Southern Ocean) yielded an RMSE of 13.79 μatm and a mean biased error of 10.52 μatm which is comparable with the Southern Ocean results for CSIR-ML6 (Gregor et al., 2019). Table 2 also shows that although all the additional high-resolution platform experiments reduced the RMSE and MBE, the magnitude of the impact was very sensitive to the platform and its location. All three scenarios of the year-long SHIP(smr) + FLOAT experiments reduced the RMSE of SHIP(smr) experiment by 32.6 - 41.9% however, only the scenario SHIP(smr) + FLOAT (PFZ) provided the lowest RMSE and MAE as well as the statistically significant correlation ($r = 0.73$) between the estimates and known truth. Both WG experiments (SAZ and PFZ deployments) also reduced the RMSE by 31.7 - 50.1% through a statistically significant correlation with $r = 0.64$ (SAZ) and $r = 0.57$ (PFZ), respectively (Table 2). The SHIP(smr) + nUSV experiment yielded the lowest RMSE (6.4 μatm)(53.5%), MAE and MBE with a significant correlation with $r = 0.74$. These results are consistent with the comparative seasonal cycle anomalies that showed the SHIP(smr) + FLOAT(PFZ) and SHIP(smr) + nUSV to have the smallest seasonal cycle biases (Fig. 6b), and higher correlations with the known truth (with $r = 0.73$ and $r = 0.74$ respectively).

## 4 Discussion

Resolving the variability and trends of the seasonal cycle of $p\mathrm{CO}_2$ in the Southern Ocean has been a long-term objective for the ocean carbon community to reduce the uncertainties and biases of the seasonal and mean annual fluxes (Bushinsky et al., 2019; Gregor et al., 2018; Lenton et al., 2006, 2013; Mongwe et al., 2018; Monteiro et al., 2015; Sutton et al., 2021; Takahashi et al., 2009). This started with largely observation-based approaches which constrained the seasonal cycle climatology (Takahashi et al., 2009, 2012) and set requirements to resolve the variability (Lenton et al., 2006; Monteiro et al., 2015). The advent of a globally coordinated surface ocean $CO_2$ data, SOCAT (Bakker et al., 2016), together with machine learning methods (Landschützer et al., 2014, 2016; Rödenbeck et al., 2015) provided a basis for spatial and temporal gap filling that has resulted in an internally consistent set of reconstructions for the ocean and Southern Ocean $CO_2$ fluxes that contribute to the global carbon budget (Canadell et al., 2021; Fay et al., 2021; Friedlingstein et al., 2021).

However, (Gregor et al., 2019) argued that the uncertainties and biases of $CO_2$ flux reconstructions are now limited by both data gaps and variability-scale sensitivity of surface ocean $CO_2$ observations – a boundary that the authors dubbed "the wall". Our results make the key point that the seasonal and mean annual biases and uncertainties (RMSEs) in the reconstructions depend critically on simultaneously resolving the spatial, meridional gradients, and temporal, seasonal and intra-seasonal variability. We now discuss three sampling scale sensitivities emerging from our analysis and what we suggest is required to get "over the wall": (1) the sensitivity of the reconstructions to the seasonal cycle, (2) the sensitivity of the reconstructions to the seasonal cycle of the meridional gradients, (3) the sensitivity of the reconstructions to the intra-seasonal variability, (4) the

need to simultaneously sample the meridional gradients and their intra-seasonal variability to get "over the wall", and (5) the limitations of this study.

## 4.1 Seasonal sampling scale sensitivity

The SHIP-only sampling experiments, which most closely simulate the historical ship-based and seasonally biased SOCAT gridded database in the Southern Ocean, point towards an unexpectedly high sensitivity of the reconstruction uncertainties and biases to the seasonal sampling scales (Figs. 1b and 5a-b, and Table 1). Simulation of the existing Southern Ocean ship-summer sampling, SHIP(smr), resulted in a seasonal cycle reconstruction with a strong positive winter out-gassing seasonal anomaly bias of ±20 µatm that was strong enough to reverse the in-gassing flux from the model domain (Fig. 5c), which also biased (positively) the spatial mean annual flux for the domain (Fig. 5a). The impact of the biased summer sampling is also expressed in the comparatively elevated RMSE: 13.79 µatm (Table 1), which is of a magnitude close to the RMSEs of the ML methods for the Southern Ocean - particularly in the Polar Frontal Zone (PFZ). For example, (Gregor et al., 2018) reported in the PFZ an average RMSE value of 14.33 µatm, and also RMSE = 13.09 µatm for the SOM-FNN method (Landschützer et al., 2016) within the same region (PFZ). Furthermore, a comparative analysis of the SHIP summer-only experiment, SHIP(smr), and the SHIP summer and winter one, SHIP(smr+wtr), shows that SHIP(smr+wtr) outperformed SHIP(smr) across all the performance metrics (Table 1) by halving them, for instance, RMSE = 6.88 µatm. The sensitivity of the reconstruction to the seasonal sampling bias is again further emphasised by the impact of the addition of a single SHIP meridional 2-leg winter (Jul-Aug) sampling lines, SHIP(smr+wtr), which reduced the mean monthly winter anomaly of $pCO_2$ for the whole domain from ±20 µatm in winter to less than 5 µatm over the whole seasonal cycle (Fig. 5a-c). The impact of the additional winter line is also expressed in the reduction of the bias error from (Table 1).

When splitting the anomalies across the two sub-domains (SAZ and PFZ) for the SHIP(smr) scenario, a comparable seasonal sampling bias sensitivity was found for the SAZ and PFZ domains (Fig. 7a). The winter reconstruction bias dominates any internal variability in the sub-domains. However, the introduction of the SHIP-winter line not only impacted on the overall mean seasonal bias but also shows that the mean seasonal cycle comprises out-of-phase seasonal modes of variability in both SAZ and PFZ domains (Fig. 7b).

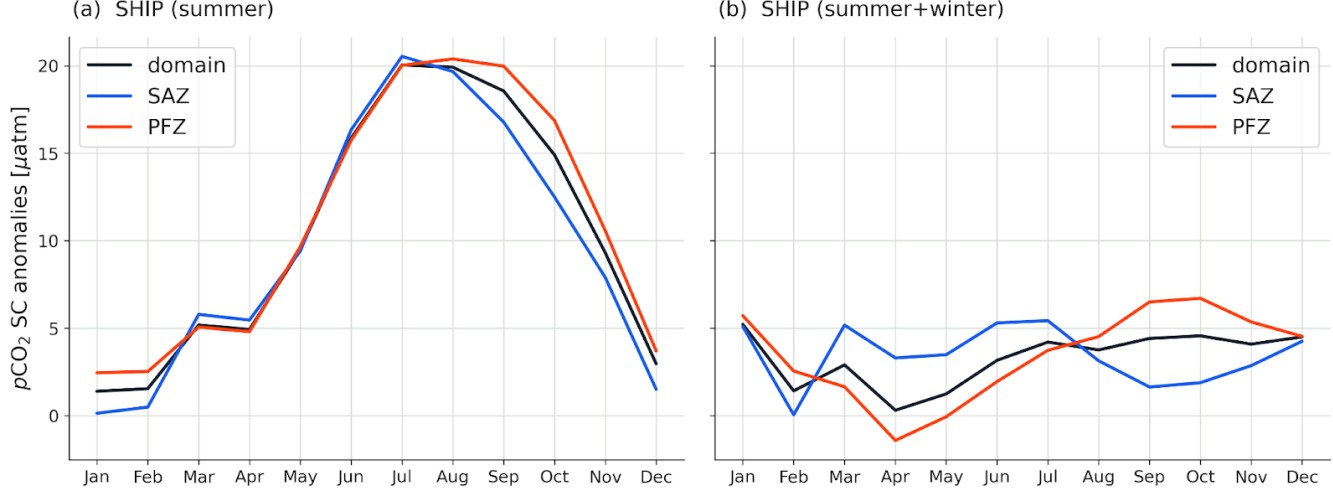

**Figure 7: Anomalies of the mean surface ocean $pCO_2$ seasonal cycle (SC) reconstructions from two SHIP-only experiments. Panel (a) shows the $pCO_2$ SC anomalies from the SHIP (summer-only)-based reconstruction in the whole domain, the SAZ, and the PFZ; and in contrast, panel (b) shows the $pCO_2$ SC anomalies from the SHIP (summer + winter)-based reconstruction for the whole domain, the SAZ, and the PFZ.**

It suggests that an important outcome of the reduction of seasonal and mean biases is the emergence of important modes of variability that can provide a useful window into key processes as well as identifying key modes of variability that can influence sampling strategies (Fig. 7a-b). Our findings on the sampling bias sensitivity are consistent with the early estimates of the minimum number of ship transects required to observationally resolve the seasonal cycle in the Southern Ocean as being quarterly, across the 4 seasons, and zonally 30º apart (Lenton et al., 2006; Monteiro et al., 2010). Together with these early results our analysis confirms that additional ship $pCO_2$ observation lines in summer will not be a useful contribution towards reducing the uncertainties and biases of the reconstructions. Rather, as proposed earlier, additional seasonal sampling lines in winter will make a decisive impact (Figs. 5a-c and 7a-b, and Table 1). However, realistically this is not achievable because access to the Southern Ocean outside the summer period is logistically challenging outside the Drake Passage (Gray et al., 2018; Monteiro et al., 2015).

The well-recognized seasonal sampling bias problem, outside the Drake Passage (Munro et al., 2015), is being addressed globally and in the Southern Ocean using a variety of autonomous sampling platforms such as Wavegliders, pH-Floats, and Saildrones (Bushinsky et al., 2019; Gray et al., 2018; Monteiro et al., 2015; Sutton et al., 2021; Williams et al., 2017). We now discuss the effectiveness of each one through experiments to simulate their sampling characteristics inside the model domain. All these experiments include the SOCAT-like SHIP-summer observations. These experiments focus primarily on the impact of the autonomous sampling platforms WG and pH-Floats as both have been deployed in the Southern Ocean with sampling strategies that view to address the seasonal sampling bias (Gray et al., 2018; Gregor et al., 2019; Monteiro et al., 2015). We return to the potential of Saildrones later in the discussion in the context of a discussion of how to "get over the wall".

## 4.2 The seasonal cycle of the meridional gradients

One of the unexpected results from our analysis was that the ship-based reconstruction with both summer and winter crossings of the domain, SHIP(smr+wtr), performed as well as the best reconstructions in which the SHIP summer-only sampling, SHIP(smr), is supplemented with an autonomous vehicle WG or FLOAT sampling continuously throughout the year (Tables 1 and 2). Thus, SHIP(smr+wtr) performed better (e.g., RMSE = 6.8 µatm) than the SHIP(smr) + FLOAT(PFZ) and SHIP(smr) + WG(SAZ) experiments that produced RMSEs of 8.0 µatm and 6.88 µatm, respectively. These results suggest that while resolving the local seasonal cycle of the surface ocean $p\mathrm{CO}_2$ with the WG and the FLOATs had a decisive impact on the RMSEs and mean biases (MBEs), an additional scale is being resolved by the SHIP experiment in winter, which is not addressed by the sampling scales of the two autonomous sampling platforms WG (1-day period) and FLOAT (10-day period). Here, we propose that the critical missing scale is the variability of the meridional gradient of surface ocean $p\mathrm{CO}_2$ (Fig. 8a), or more critically, the seasonal cycle of the meridional gradient of $p\mathrm{CO}_2$ (Fig. 8b). Together these figures highlight that although the mean increasing southward gradient in $p\mathrm{CO}_2$ is sustained throughout the annual cycle (Fig. 8a), there are sharp seasonal spatial and temporal contrasts in the meridional variability of the magnitudes (Fig. 8b). This includes significant seasonal differences in the influence of mesoscale on the spatial variability (Fig. 8a). The climatological meridional gradients of surface DIC and $p\mathrm{CO}_2$ in the Southern Ocean are well characterized through *in situ* observations (Wu et al., 2019), data products (Gregor et al., 2018, 2019) and models (Hauck et al., 2015, 2020). These results highlight that characterizing the meridional gradient is not sufficient in itself because shipboard observations in the SOCAT database already include the meridional gradients but these observations in the Southern Ocean are strongly biased toward summer (Gregor et al., 2019; Gregor and Gruber, 2021). As our study indicates, the seasonal scale variability of that meridional gradient matters the most, which is why SHIP(smr+wtr) makes such a difference (Tables 1 and 2) compared to SHIP(smr) + WG and SHIP(smr) + FLOAT.

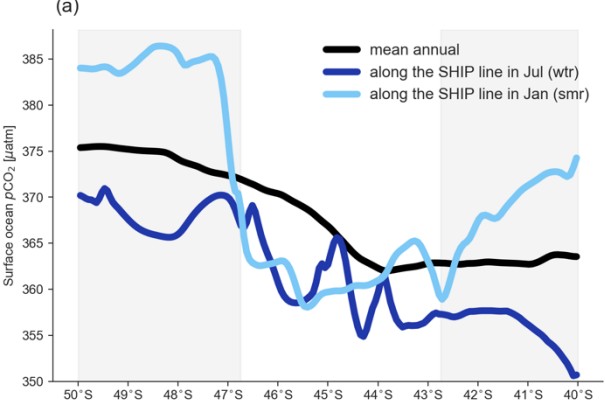
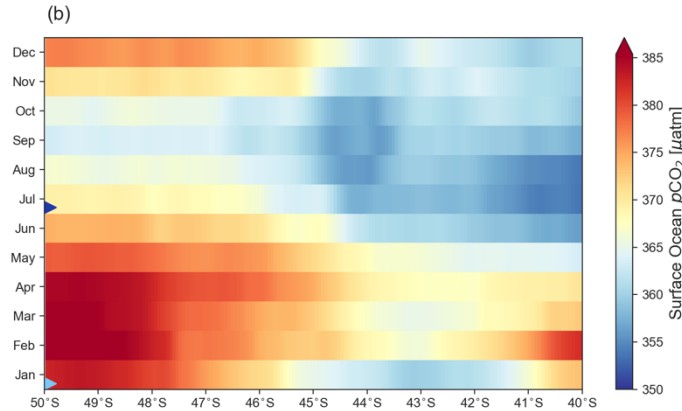

**Figure 8: Seasonal contrasts for the meridional gradient (MG) of surface ocean $pCO_2$ in the experimental sub-domain. Panel (a), shows the mean annual MG (black), the mean MG along the SHIP line in summer (January)(light blue) and in winter (July)(dark blue); and panel (b) shows the seasonal cycle of the meridional gradient of $pCO_2$ showing the months when the SHIP sampled (blue triangle markers) with the light blue for Jan (smr) and the dark blue for Jul (wtr). The light grey shadings in panel (a) show the sub-domain areas (north and south) where large differences in $pCO_2$ meridional gradients along the SHIP line in summer and winter.**

Significant differences exist between the meridional gradients along the SHIP line in summer (e.g., January) and winter (e.g., July) (Fig. 8a-b). For example, these differences are more significant farthest south (> 47ºS) and farthest north (< 43ºS) compared to the middle (43ºS - 47ºS) of the sub-domain (light grey shadings, Figs. 8a). Similarly, the seasonal cycle difference is not as big in the middle of the sub-domain as it is at the extreme lines of the SAZ and PFZ (Fig. 8b). That is why we need a sampling platform that is able to capture critical scales of variability. Another key point we raised concerning the sampling scale sensitivity on the $pCO_2$ reconstructions is that resulting uncertainties and biases depend on the seasonal scale of the meridional gradients of the surface ocean $pCO_2$ (Fig. 8b). Shedding light of this point results in resolving the seasonal cycle of the meridional gradients.

The similarity of the anomalies between the SHIP(smr) + WG(SAZ) and SHIP(smr) + nUSV experiments are supported by the impact that these sampling strategies have on the seasonal cycle of the bias (Fig. 6b). This shows that, relative to other sampling experiments, there was a reduction of the biases across the whole seasonal cycle but more so in summer-autumn and less so in winter-spring (Fig. 6b). The significantly smaller MBE for SHIP(smr) + WG(SAZ) can be ascribed to the bias being slightly negative in summer-autumn and positive in winter-spring which leads to a small mean annual MBE whereas in the case of the SHIP(smr) + WG(PFZ) experiment, the MBE is small but positive throughout (Fig. 6b, and Table 2). The mean annual anomaly map of $pCO_2$ for the SHIP(smr) + nUSV experiment still shows a positive anomaly, though weaker, at the frontal zone because although the nUSV Saildrone has a daily sampling resolution, it is only crossing the highly synoptic SAF zone periodically (Fig. 1c). This is consistent with all the instances when not resolving the temporal variability results in a positive bias of varying magnitudes (Fig. 6b).

On designing an observation-based strategy for quantifying the Southern Ocean uptake of CO₂, (Lenton et al., 2006) argued that constraining the net seasonal air-sea CO₂ fluxes within the natural variability of the carbonate system requires doubling the current Southern Ocean meridional sampling. In a semi-idealized experimental setting, our study takes this further by showing that resolving the seasonal cycle of the meridional gradients is very critical. WG and FLOAT provide high temporal sampling resolution, but they do not resolve the existing meridional gradients. Therefore, increasing data density through zonal autonomous sampling vehicles (e.g., floats) is not sufficient to minimize reconstruction errors. The quarterly meridional sampling strategies proposed by (Lenton et al., 2006) and (Monteiro et al., 2010) could help to resolve the seasonal cycle of the meridional gradients, but they are not operationally feasible.

## 4.3 Intra-seasonal variability of the Seasonal Cycle

Recent high-resolution observations using different types of carbon-enabled autonomous platforms have highlighted a potential sensitivity of Southern Ocean $CO_2$ flux reconstruction uncertainties and mean bias to aliases in sampling the intra-seasonal to seasonal temporal scales (Bushinsky et al., 2019; Gray et al., 2018; Monteiro et al., 2015; Sutton et al., 2021; Williams et al., 2017). Here we discuss the sensitivity of the model domain reconstruction statistical metrics to a range of semi-idealized scenarios of SHIP-summer supplemented with FLOAT and WG observations (Table 2; Fig. 6). In each case of the

FLOAT and WG, they were made to sample each sub-domain (SAZ and PFZ) for a year at their characteristic sampling periods of 10 days and 1 day, respectively. The assumption was that the FLOAT would remain in the domain throughout the year. Thus, to not disadvantage the floats in these experiments, one float was deployed in each sub-domain (SAZ and PFZ) as shown in Fig. 1b, under the assumption that floats would not cross the Sub-Antarctic Front (SAF). The nUSV, Saildrone analogue, sampling scenario is brought in later to test the predicted sampling requirements to achieve the lowest RMSEs and mean bias

error. There was no real benefit in reproducing the zonal sampling approach for the Saildrone (Sutton et al., 2021) because it would be comparable to the zonal travel of FLOAT but with higher daily sampling more akin to the WG. Its metrics would therefore have been comparable to both and contributed little to learning.

One of the standout aspects of this part of the analysis, investigating the impact of the sampling period, was the significant

difference in the uncertainty and biases between the best performing SHIP(smr) + WG(SAZ) (RMSE = 6.88 µatm; MBE = 0.82 µatm) and SHIP(smr) + FLOAT(PFZ) (RMSE = 8µatm; MBE = 5.32 µatm) scenarios (Table 2). These comparative statistics point to the reconstructions also being very sensitive, particularly to the temporal sampling scales. This finding can be explained and understood from the characteristics of the variability from time series from single model grid cells in the SAZ, on the SAF, and in the PFZ (Fig. 9). Local scale single grid-cell observations are appropriate instead of spatial means

because they simulate the local nature of the variability and how it is observed. The variability characteristics of these time series help explain the statistics of the $pCO_2$ reconstructions (Fig. 9; Table 2). The SAZ and SAF are characterized by stronger intra-seasonal variability whereas the PFZ is characterized by lower frequency (sub-seasonal) – seasonal modes of variability (Fig. 9). Thus, while the SAZ and SAF sub-domains and their stronger intra-seasonal variability are best resolved by the daily sampling of the WG, the PFZ domain, which is dominated by the lower frequency sub-seasonal to seasonal cycle, is resolved

equally well by the WG – daily and FLOAT – 10-daily sampling periods (Fig. 9; Table 2).

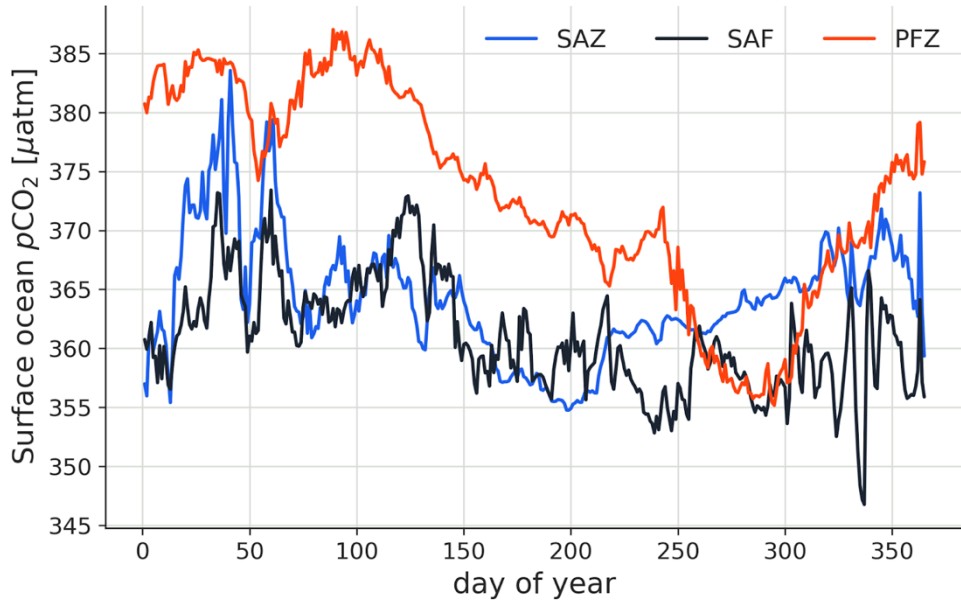

**Figure 9: Time series (one year) plots of the variability of surface ocean $pCO_2$ at single model grid cells on the SHIP line (2.5ºE, Fig. 1b). We used the following single model grid cells: 42ºS, 2.5ºE in the Sub-Antarctic Zone (SAZ); 44ºS, 2.5ºE on the Sub-Antarctic Front (SAF); and 47ºS, 2.5ºE in the Polar Frontal Zone (PFZ). It shows that while the SAZ and SAF are dominated by synoptic modes of variability, the PFZ is characterized by longer period sub-seasonal to seasonal scales of variability.**

Therefore, given that WGs and FLOATs in these sampling scenarios are comparable in that neither have a strong meridional gradient resolving sampling strategy, the main difference between them is the daily sampling rate of the WGs and the 10-day sampling rate for the FLOATs. Fig. 9 then helps explain why even though the domain reconstructions based on the FLOAT(PFZ) sampling scenario perform best out of the two FLOAT scenarios, SHIP(smr) + FLOAT(SAZ+PFZ), ultimately it underperformed relative to the WGs because it was aliasing the synoptic intra-seasonal variability in the SAZ and SAF. This surprising performance of SHIP(smr) + FLOAT(SAZ+PFZ) after running the experiment several times likely resulted from the difference in modes of variability in the SAZ and PFZ (Fig. 9). The float did well when deployed in PFZ dominated by seasonal variability which can be resolved by the 10-day sampling period but performed poorly when it was deployed in the SAZ characterized by intra-seasonal modes which cannot be resolved by the 10-day sampling period. Thus, when sampling the two sub-domains simultaneously, SHIP(smr) + FLOAT(SAZ+PFZ) resulted in a poorer performance than for the PFZ alone (Fig. 6b; Table 2). The finding that the high temporal resolution of the SHIP(smr) + WG(SAZ) was the only sampling combination to match the performance of the SHIP(smr+wtr) experiment whose strength was in resolving the seasonal contrasts of the spatial meridional gradient, suggests that these two scales of variability, intra-seasonal and meridional, are close to equally important towards achieving a low bias and RMSE reconstruction. Resolving the former and the latter simultaneously may therefore be a presently missing critical step.

More broadly and relative to the SHIP summer-only scenario, all the annual cycle experiments yielded a reduction in the reconstructed seasonal cycle anomalies (Fig. 6b) and in the uncertainties (32 – 50%), biases (±50%) as well as a statistically significant improvement for Pearson's correlation coefficient ($r$) (Fig. 6a-b, and Table 2). When comparing SHIP(smr) + WG with SHIP(smr) + FLOAT, reconstructed annual mean $pCO_2$ maps for the whole domain were consistent with reduced anomalies, for instance, with small positive anomalies for SHIP(smr) + FLOAT(PFZ) and small negative anomalies for SHIP(smr) + WG(SAZ) (Fig. 6d-e, respectively). However, while comparing SHIP(smr) + WG(SAZ) with SHIP(smr) + FLOAT(SAZ) where the WG and FLOAT are both deployed in the SAZ, there is a significant difference in the RMSEs and MBEs with respectively 6.88 µatm and 0.82 µatm for the former, and 9.29 µatm and -4.81 µatm for the latter (Table 2).

This analysis provides additional understanding of the strengths and limitations of the way that the 3 main autonomous platforms (Wavegliders, carbon-floats and Saildrones) deployed in the Southern Ocean contribute to increasing or decreasing the seasonal cycle and mean annual biases as well as the RMSEs (Monteiro et al., 2015; Bushinsky et al., 2019; Sutton et al., 2021). Based on hourly observations of the surface ocean $pCO_2$, Monteiro et al. (2015) showed that a temporal sampling resolution of less than 2 days would be necessary in 30 – 40% of the Southern Ocean, corresponding to the SAZ, to reduce the uncertainty to less than 10% of the annual mean (Sect. S3.5). Our study confirms the sensitivity of the RMSE of the intra-seasonal variability sampling alias and also shows its impacts on the bias of the annual mean. SOCCOM-float calculated $pCO_2$ data has made a decisive impact on resolving the seasonal cycle in the Southern Ocean and suggests that winter CO₂ out-gassing may be underestimated in SOCAT-based reconstructions (Bushinsky et al., 2019; Gray et al., 2018). Our study suggests that these observed and reconstructed elevated out-gassing fluxes may be the result of both aliasing of the intra-seasonal variability as well as not resolving the seasonal cycle of the meridional gradient. Our analysis also raises a question around the assumption that not resolving the intra-seasonal variability of $pCO_2$ does not contribute significantly to the RMSE and the bias (Bushinsky et al., 2019). It shows that the intra-seasonal modes of the wind are not sufficient to impart a low mean annual and seasonal cycle bias.

To provide a more quantitative characterization of our findings, an additional analysis was conducted on the sub-10-day mode of variability. A 10-day rolling mean was used to eliminate or weaken the sub-10-day mode of variability (Fig. S8a). The difference between this 10-day rolling mean and the daily model output gives the high-frequency variability and the root-mean-squared error (RMSE) gives us a statistical understanding of what the uncertainty might be if we sampled at a 10-day rate (shown in Fig. S8b as a map). The resulting mean RMSEs for the SAZ and PFZ, after implementing the 10-day rolling mean, are 2.53 µatm and 1.71 µatm respectively, a significant reduction relative to the RMSEs for the FLOAT experiment using the daily model output (Table 2). This provides further quantitative support for our findings and the work of Monteiro et al. (2015) that more dynamic regions require higher sampling rates. We finally propose that the impact of SOCCOM floats on the reconstructions can be strengthened by reducing the sampling period to < 2 days, especially in high EKE areas, and through a coordinated meridional deployment strategy that helps to resolve the meridional gradient across the annual cycle.

Our study also suggests that notwithstanding the high temporal frequency of the USV Saildrone, the present emphasis on a zonal sampling pattern (Sutton et al., 2021) also underestimates the potential contribution that this platform could make in observing the seasonal cycle of the meridional gradient at high temporal resolution simultaneously. We now examine this aspect in more detail.

## 4.4 "Getting over the wall" in the Southern Ocean by simultaneously resolving the intra-seasonal and seasonal variability of the meridional gradient – Proposed optimal sampling strategy

This analysis has highlighted that in order to minimize the uncertainties and biases sufficiently to "get over the wall", observational strategies in the Southern Ocean need to simultaneously resolve the seasonal cycle of the meridional gradient at temporal scales that also resolve, where necessary, the intra-seasonal variability. To test this hypothesis, we designed an additional year-round Observing System Simulation Experiment (OSSE) that simulated the spatial and temporal sampling capabilities of the new unmanned surface vehicle (nUSV) Saildrone (Sutton et al., 2021) to supplement the SHIP summer-only sampling SHIP(smr) (Figs. 1c and 6b, h); that is, SHIP(smr) + nUSV. This experiment combined the speed of the nUSV Saildrone (Gentemann et al., 2020; Meinig et al., 2019) required to cover the regional meridional spatial gradients length scales (Fig. 1c), with high-frequency daily sampling to supplement SHIP(smr). Together these fulfill the requirements that emerged from the earlier analysis.

Comparative statistics show that the SHIP(smr) + nUSV experiment yielded a very significant improvement in the reconstruction skills relative to all other platform combinations (Table 2). Its performance metrics (RMSE = 6.4 µatm) outperformed the next best combination SHIP(smr) + WG(SAZ) (RMSE = 6.88 µatm) and SHIP(smr) + FLOAT(PFZ) (RMSE = 8.0 µatm). This supports the hypothesis that resolving the intra-seasonal and seasonal variabilities of the meridional gradients is decisive in minimizing uncertainties and bias in $pCO_2$ reconstructions. Based on this analysis, we propose that the optimal sampling scheme is the SHIP + nUSV because it provides not only a high temporal resolution (daily) of the large-scale meridional gradients but also combines speed to cover the required meridional spatial extent.

The nUSV Saildrones are still relatively new autonomous sampling platforms and their ability to withstand the stringent weather and sea conditions in the Southern Ocean are still being assessed (Sutton et al., 2021). Recent deployments of Saildrones have been focused on zonal circumpolar tracks, which have been successful in proving the Saildrones as a robust sampling platform, and in observing the seasonal cycle of $CO_2$ fluxes in the sub-polar domain (Sutton et al., 2021). This approach is comparable to the zonal sampling of FLOATs (Fig. 1b) but with a higher temporal sampling frequency (daily vs 10-day). Notwithstanding the higher temporal sampling frequency from the Saildrone, the lack of a meridional spatial component to the zonal sampling strategy limits its value in reducing the uncertainties and biases of any reconstructions that use them. Its inclusion in $CO_2$ flux reconstructions would improve the RMSE and mean bias error (MBE) relative to SOCAT-

based reconstructions which, as discussed earlier, is not where autonomous sampling vehicles can add the best value (Tables 1 and 2).

Our work here shows that a zonal sampling strategy, while good for operational navigational reasons, is not the most efficient
way to maximize the value of USV Saildrones sampling to resolve critical scales of variability necessary for high confidence in the $pCO_2$, and inferred $CO_2$ flux reconstructions in the Southern Ocean. Furthermore, our study shows how by mixing the meridional sampling strategy (Lenton et al., 2006; Monteiro et al., 2010) with the current zonal sampling we can leverage the USV Saildrones to make sure we are not missing the meridional gradients.

**4.5 Applicability of the sub-domain to a wider Southern Ocean**

The focus of this study was on investigating the mismatch between sampling periods and the modes of variability of pCO2 in the domain rather than the mechanisms. This selected domain in the South-East (SE) Atlantic Ocean encapsulates the contrasts in the scales of variability of interest, namely the seasonal and intra-seasonal modes that are characteristics of the Southern Ocean (Fig. 10). It shows how findings in the study domain can be extended to the Southern Ocean. Using a 10-year period of
pCO2 output from NEMO-PISCES model simulations at a 5-day temporal mean, the Seasonal Cycle Reproducibility (SCR) of pCO2 was calculated as the correlation of the detrended pCO2 with its own 10-year climatology – the larger the correlation, the stronger the SCR (Thomalla et al., 2011). This resulted in the SCR-based clustering of the Southern Ocean into three regions (Fig. 10) corresponding to the low (LSCR), medium (MSCR), and high (HSCR) SCR areas, respectively. The criteria of the choice of these three ranges are as follows. In high SCR areas, there is no intra-seasonal variability and annual signals.
In medium SCR areas, intra-seasonal variability emerges but is smaller in magnitude compared to the seasonal cycle, while in low SCR areas, there is no seasonal signal, and the intra-seasonal variability is larger than the seasonal cycle.

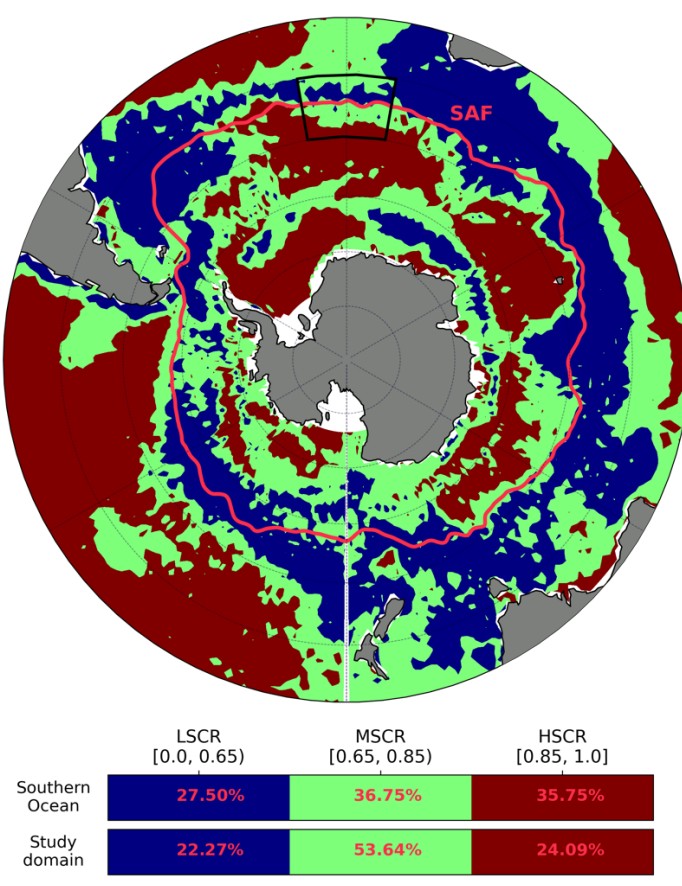

| | LSCR [0.0, 0.65) | MSCR [0.65, 0.85) | HSCR [0.85, 1.0] |
|---|---|---|---|
| Southern Ocean | 27.50% | 36.75% | 35.75% |
| Study domain | 22.27% | 53.64% | 24.09% |

**Figure 10: Map showing the study domain and the Southern Ocean sub-regions resulting from the seasonal cycle reproducibility (SCR) of pCO2 calculated based on 10 years of NEMO-PISCES simulations at the 5-day temporal resolution, where the Sub-Antarctic Front (SAF) (light red) and the study domain (black box) are depicted. The table below the map shows the fraction coverage estimates (%) for these SCR-based regions both in the domain and in the Southern Ocean as a whole. LSCR corresponds to low SCR areas, while MSCR and HSCR respectively represent medium and high SCR areas.**

Although this study domain was chosen within a high-EKE area (black box; Fig. 1a; Fig. 10) because of its contrasting seasonal and intra-seasonal variability of the surface ocean $pCO_2$, the SCR metric shows how the study area in the SE Atlantic Ocean contrasts to the Southern Ocean as a whole (Fig. 10). As argued in the previous paragraph, seasonal and intra-seasonal variability is relatively associated with LSCR (0-0.65) and MSCR (0.65-0.85) regions, which together represent ~75% of the study domain, and ~64% of the whole Southern Ocean (cf. table shown in Fig. 10). This demonstrates that the subdomain modes of variability (which are dominantly intra-seasonal) may be applied to the wider Southern Ocean.

Longitudinally, the Southern Ocean is equal to 360º/20º = 18 times our 20ºW-E domain. However, while in theory, our domain is 1/18[th] of the zonal extent of the Southern Ocean, it represents different modes of variability as argued above. Thus, we should be able to capture the variability with less than 18 USV Saildrones. Based on this, we have a speculative estimate of

the monetary cost, see the Supplementary Materials (Sect. S3.4). A study on the full Southern Ocean will be performed to assess this more thoroughly.

### 4.6 Limitations of the study

In this study, our limitations were tied around three main points: the model used, the selected sub-domain, the existing shift in the seasonal cycle phasing of the model and data products, and the overfitting tendency of ML models. Here we discuss these limitations separately.

We only had one year of daily outputs of the high-resolution coupled (NEMO-PISCES) ocean model, BIOPERIANT12 (BP12). These BP12 model spatial (1/12º by 1/12º) and temporal (daily) resolutions influenced the designing of the OSSEs, therefore impacting the sampling approach of the synthetic platforms compared to their real-world counterparts. For example, unlike other sampling platforms that can be driven remotely, floats are harder to simulate due to the way they operate. Thus, we could only mimic the 10-day sampling period and the deployment location and assume that they are randomly transported eastward by the water current. Since the Antarctic Circumpolar Current (ACC) moves eastward the random walk we implemented is an adequate approximation and adds an element of stochasticity that is likely close to reality (Fig. 1b).

The selected sub-domain combines regional and mesoscale gradients and features (such as eddies and fronts) which could challenge the reconstruction methods to better capture some variability scales such as the seasonal cycle of the meridional gradients (Fig. 8). However, the meridional gradients could be also associated with the meandering of the ACC fronts such as the SAF that crosses the domain. On the other hand, the assumption of the domain representativeness of the variability scales of the region could be a cause of concern as this would be applicable in regions where latitudinal gradients are strong. For example, the BP12 model output might not achieve this assumption based on a standard deviation of 9.1 µatm for the synthetic SHIP data compared to 20.96 µatm of SOCAT data in the sub-domain.

Existing differences in the mean $p\mathrm{CO}_2$ seasonal cycles of the model and data products (Fig. 2) could also result from processes that deterministic models such as the BP12 ocean model (NEMO-PISCES) cannot yet constrain, due to a lack of understanding of the complete Southern Ocean carbonate system or mixed layer physics (Lenton et al., 2013; Mongwe et al., 2016; Monteiro et al., 2015). However, our knowledge of which one is right between model and data products remains limited.

Lastly, overfitting is a common challenge in supervised machine learning (ML) problems. Although each of the two ML algorithms (FNN and GBM, see Sect. 2.4) best practices were in training, the GBM algorithm encountered more challenges with the overfitting compared to the FNN (cf. Table S3). While GBM has been proven to deal well with imbalanced or sparse

datasets (Ke et al., 2017), it is more likely to overfit the training data because of the model's potential for high complexity (Frery et al., 2017).

Finally, while studies such as Gregor et al. (2019); Devil-Sommer et al. (2019); Gloege et al. (2021) found that mixed layer
depth (MLD) climatology is an important predictor of surface ocean pCO2, our use of dynamic model generated MLD may impart some advantage, which might not be available to the real-world observation-based reconstructions. Moreover, we also recognize that model-generated Chl-a may not be, in absolute terms, directly analogous to satellite Chl-a. However, these advantages from using model output are uniform across all the sampling experiments in this study.

**5 Conclusions**

From this study, we propose that one can advance the uncertainties and biases from machine-learning $pCO_2$ reconstructions "beyond the wall", at least in the Southern Ocean. Within a chosen experimental domain of the Southern Ocean, we demonstrate that this would require resolving the seasonal and intra-seasonal modes of variability of the meridional gradients of $pCO_2$ through a combination of high frequency (at least daily) observations spanning the meridional axis. We showed that
the reconstructed seasonal cycle anomaly and mean annual $pCO_2$ are highly sensitive to seasonal sampling biases. The seasonal sampling bias comprises both the temporal and meridional spatial scales of variability. This may explain the significant winter-positive bias in the reconstruction of the seasonal cycle of $pCO_2$ in the domain, which likely may also be contributing to the apparent winter-maximum outgassing or weakening of the ingassing of $CO_2$ observed in recent Southern Ocean data products. This points to an urgent need to address the existing seasonal bias (towards summer) in the Southern Ocean SOCAT dataset
through improving the sampling strategy of the present autonomous platforms, so they are better aligned to the integrated spatial and temporal sampling scale needs.

Inside the chosen domain, the study confirmed that not resolving the high frequency (synoptic - sub-seasonal) variability results in insufficient decreases in mean biases and RMSE scores for the reconstructed mean annual flux. Present 10-day sampling
periods of floats have a limited impact on reducing uncertainties and biases in $pCO_2$ mappings because they do not resolve the intra-seasonal variability. In addition, the predominantly zonal and quasi-Lagrangian sampling does not contribute sufficiently to resolving the seasonal variability of the meridional gradients of $pCO_2$. Our study proposes that a more meridionally coordinated deployment of floats could contribute further to resolving synoptic variability and the meridional gradients. For example, increasing sampling frequency to < 2 days, particularly in high-EKE areas as well as a meridionally coherent
sampling strategy would support resolving the synoptic-scale variability and the variability of the basin-scale gradients. Although they still lack the meridional gradient reach, Wavegliders in pseudo-mooring modes improve on floats (RMSEs, MBEs) and the main explanation for this improvement is because of their higher sampling frequency (daily). This study

recommends that the use of Wavegliders in the reconstruction of $CO_2$ fluxes in the pseudo-mooring mode should be discontinued and adopt a meridional dimension to the high temporal resolution (1-2 days). We showed that while the USV Saildrones in the present zonal sampling mode improve the RMSEs and biases, this might not be the most efficient way to maximize their strengths stemming from their high sampling frequency (hourly) and large spatial scale (by leveraging their speed). We thus propose that USV Saildrones are probably the optimal platforms to address the necessary integrated large-scale spatial and high-resolution temporal sampling.

In summary, ship-based observations (SOCAT-like) remain vital to the reconstruction of $CO_2$ fluxes in the Southern Ocean as a whole and should be continued. These observations are the baseline data involved in the training of any machine learning algorithms behind the main observation-based products of reference. However, these ship-based observations are seasonally biased (towards summer) due to under-sampling during stormy autumn and winter seasons, which are likely the root of persistently elevated uncertainties and a winter-positive bias in the reconstructions. This bias should be addressed with urgency. Finally, this study proposes that a meridional sampling strategy may be an efficient way of sampling using autonomous observing systems. In this case, we recommend that existing ship-based observations of the surface ocean $pCO_2$ in the Southern Ocean should be supplemented by year-round autonomous high-resolution observations that resolve the seasonal cycle of the meridional gradients of the surface ocean $pCO_2$. However, a follow-up study is also recommended to test, for example, the USV Saildrone effectiveness and impact on reducing uncertainties and biases of the seasonal and mean annual reconstruction of $CO_2$ fluxes in the Southern Ocean as a whole.

**Code and data availability**

Supporting codes/scripts used for data analysis are contained in the GitHub repository https://github.com/Djeutsch/SOCCO-OSSE-v1. Data used in this study have been published in the online open-source repository Zenodo and can be accessed at https://doi.org/10.5281/zenodo.5788736 (Djeutchouang et al., 2021).

**Supplement**

## Author contributions

LMD is the lead author and developed the method and wrote the manuscript. LMD and PMSM conceived the study and performed the analysis. NC set up and ran the high-resolution BIOPERIANT12 model used in the study. LG contributed to the development of the method and to editing the manuscript. MV contributed to the initial conceptualisation of the methods and proofread the manuscript. PMSM contributed substantially to the development of the manuscript and its reviews.

## Competing interests

The authors have no competing interests.

## Acknowledgements

This work is part of a PhD, and the study was supported by funding from the Department of Science and Innovation (DSI), South Africa, the National Research Foundation NRF-SANAP grants SNA170522231782, SNA170524232726 and CSIR Parliamentary Grant. We are grateful for the technical support and computational hours from the Centre for High Performance Computing (CSIR-CHPC). LG acknowledges funding from ETH Zürich and the European Commission through the COMFORT (grant no. 820989).

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
