# Peer review of "The sensitivity of $p\text{CO}_2$ reconstructions to sampling scales across a Southern Ocean sub-domain: a semi-idealized ocean sampling simulation approach"

_Biogeosciences, 2021_

## Author Comment (AC1)

*Response to reviewer 1 of "**The sensitivity of pCO2 reconstructions in the Southern Ocean to sampling scales: a semi-idealized model sampling and reconstruction Approach**"*
*by Djeutchouang, Chang, Gregor, Vichi, Monteiro*

*First of all, we would like to take the opportunity to thank the reviewer for the thoughtful comments and suggestions, which help strengthen the paper. We will respond (in italics) to each of your specific comments as follows.*

**This paper is well-written, and some interesting conclusions can be drawn from the results.**

*Thank you*

**However, I had trouble appreciating the paper as the conclusions were presented in what felt to be a very misleading way. Recommendations are given in the abstract and conclusions without emphasizing appropriate caveats making this paper feel biased and unscientific. I recommend major revisions to appropriately implement these caveats. The title and abstract are misleading with the conclusions not put in proper context. I would prefer if you took Southern Ocean out of the title. As it stands the paper tells me much less about the Southern Ocean than I had envisioned it would. Your decisions made in this paper are fine, but they significantly impact the results and cannot be hidden in the details. Indeed the experiment was designed to make nUSV do the best as you explain on line ~657. Here are some more specific comments (in no real particular order):**

*Response: We thank the reviewer for these comments and suggestions. We have revised the title of the study as follows: "**A semi-idealized model sampling and reconstruction approach across a Southern Ocean front: the sensitivity of pCO2 reconstructions to sampling scales**".*

*After revising the manuscript, we have emphasized appropriate caveats as suggested and the abstract has been revised as follows: "... Here we choose one year of an experimental model sub-domain of 10 degrees of latitude (40ºS - 50ºS) by 20 degrees of longitude (10ºW - 10ºE). This domain is divided by the Sub-Antarctic Front (SAF) and thus overlaps both the Sub-Antarctic Zone (SAZ) and Polar Frontal Zone (PFZ) which are the two most sampled sub-regions of the Southern Ocean. While our study domain does not resolve all Southern Ocean scales, it is representative of the scale variability we aim to address. The OSSEs simulated within this domain the spatio-temporal sampling scales of surface ocean pCO2 in ways that are comparable to ocean CO2 observing platforms (Ship, Waveglider, Carbon-float, Saildrone) in terms of their temporal sampling scales, and not necessarily their spatial ones. This study focuses more on the mode of variability and not the mechanisms. ... From our study, it appears that a more coordinated deployment of floats would contribute further to resolving synoptic variabilities and the meridional gradients."*

*Regarding the conclusions, they have been appropriately revised by emphasizing caveats.*

***Major comments:***

**- The choice to reconstruct one year of a small box (10 degrees of lat by 20 degrees of long) and call that the Southern Ocean is quite a leap and should be presented clearly in the abstract so as**

**to not mislead about results. This isn't the Southern Ocean and this choice has significant implications on observing system recommendations.**

*Response: We thank the reviewer for this suggestion. We do acknowledge that the chosen domain is not the Southern Ocean. This has been revised and presented clearly in the abstract. However, we recall that while our selected domain does not resolve all Southern Ocean scales, it is representative of the scale variability we aimed to address. For example, we correlated the detrended pCO2 data from three distinctly selected grid cells (black dot, Fig. 1a-c, of this note) of our study domain with the entire Southern Ocean. In each case, the analysis resulted in a high correlation (>=0.5) area covering more than 50% of the total area. Therefore, the results from this study could be representative of the broader Southern Ocean. Moreover, with respect to longitude, our domain covers about +5% of the Southern Ocean, which may be longitudinally applicable to the Southern Ocean as a whole (Lenton et al., 2006; Mazloff et al., 2018; Wu et al., 2019). However, this will still need a follow-up study.*

*We are using the model output data for this larger domain (Southern Ocean) for future work.*

[Figure]

***Figure 1:*** *Correlation (Pearson r-value) between the pCO2 from three distinct points (black dot): panels **(a)**, **(b)**, and **(b)** of our study domain (black box) and the rest of the Southern Ocean. This is quite similar to the seasonal cycle reproducibility in some way except that in this case, we are relating our domain specifically to the Southern Ocean.*

*Additionally, here, we provide another context of why we assumed the study domain (black box, Fig. 2a, of this note) is representative of the scale variability we aimed to address. Based on the 10-year model (NEMO-PISCES) data of 5-day resolution that we have, we calculated the seasonal cycle reproducibility (SCR) of the model pCO2, which is defined as the correlation of detrended pCO2 with its own climatology – the larger the correlation, the stronger the SCR (Thomalla et al., 2011). Based on this SCR, we could classify the Southern Ocean into three regions LSCR, MSCR, and HSCR (Fig. 2a, of this note) corresponding to low, medium, and high SCR, respectively. The fraction coverage estimates of these SCR-based regions are shown in Fig. (2b) (of this note). Our study domain is a high-EKE area, and increased intra-seasonal variability of pCO2 in the Southern Ocean is associated with high-EKE areas (Monteiro et al., 2015; du Plessis et al., 2019). These high-EKE areas correspond relatively to MSCR (0.65-0.85) and HSCR (0.85-1) regions which, based on our estimation, represent more than 71% of the whole Southern Ocean.*

*The data we used for this SCR calculation is from a 10-year period of 5-day resolution ocean (NEMO-PISCES) model simulations that we have.*

[Figure]

***Figure 1:*** *Panel **(a)** is the Southern Ocean regions based on the seasonal cycle reproducibility (SCR) of the model pCO2 with the Sub-Antarctic Front (SAF) (light red) and the study domain (black box), and panel **(b)** is their fraction coverage estimates (%). LSCR corresponds to low SCR areas, while MSCR represents medium SCR areas and HSCR corresponds to high SCR areas.*

**- As you state, Figure 2c shows this model is inconsistent with both reconstructions. Line ~195 needs to be changed. This is significant as it means the pCO2 may be driven by a mechanism different from the real world. This is relevant to the objective of the study (i.e. you can predict largely from knowledge of SST) and you need to discuss this. If the correlation to SST was weaker you would need to know the biological component better and the statistics may be different. This also has significant implications on observing system recommendations.**

*Response: Thank you. Regarding this comment, it is relevant to start by recalling that this study is more about the mode of variability and not the mechanisms. The forced coupled ocean model (NEMO-PISCES) does represent the processes that regulate CO2 but for the purpose of this study, it does not really matter whether the model is right or wrong. What matters is that it shows a domain that has different modes of variability, and what we tried to show is that the reconstruction is sensitive to how you sample such a domain with different modes of variability.*

*Further, the machine learning (ML) methods used in this study; that is, the feed-forward neural network, and the tree-based method gradient boosting machine, do not capture the mechanisms that actually drive the pCO2 (Holder and Gnanadesikan, 2021). Rather, these ML models capture changes in the drivers and then the associated changes in the pCO2. We do acknowledge that the lack of better knowledge of biological components might also have implications on observing system recommendations.*

**- Is nUSV sampling realistic? How many platforms are envisioned? It seems very dense – how long would it take one saildrone to reach all those points? Shouldn't the nUSV sampling look more like the ship track? Moreover, how many nUSVs would it take to sample the full Southern Ocean with that density?**

*Response: Thank you for these questions. Our starting reference was the nUSV used in Sutton et al. (2021) that sampled at a very high resolution and completed in about 6 months the first autonomous circumnavigation of Antarctica providing hourly observations. Thus, our answer to the first two questions is affirmative; that is, the nUSV sampling is realistic with one platform given the size of the sampling domain. The study is actually proposing that this meridional sampling approach may be an efficient way of sampling using autonomous observing systems.*

*Regarding the third question, we created a subset of Sutton et al. (2021)-USV dataset within the same domain of 20 degrees longitude (10°W - 10°E) as that of the study (see Fig. 3, of this note). We found that the Sutton et al. (2021) USV would take ~16 days to cover our 20°W-E domain; that is, 16days \* 24hrs = 384 hourly samples as shown in Fig. (3). However, the nUSV sampling pattern is idealised, with the goal of sampling on both sides of the front. In future work we will investigate a more realistic sampling pattern with an asociated "cruise-track". In this more realistic sampling strategy, an along-front (West to East) zig-zag pattern would be sampled across the front. This sampling pattern would likely be even better at capturing the meridional gradient across the front than the idealised sampling that we currently use. Therefore, using a back of the envelope approach, we found that the Saildrone would be able to cover our domain in 45 days using a zig-zag pattern - assuming 42°S to 46°S with each pass covering 2.5°W-E for each pass (~500 km) with 8 passess in our domain (4000 km). At a speed of 2 knots (3.7 km/hr), the drone would cover the domain in 45 days.*

[Figure]

**Figure 3:** *Sampling tracks of Sutton et al. (2021)-like USV inside our study domain.*

*About the question regarding the nUSV sampling tracks, the answer is affirmative. Actually, nUSV could sample like a ship but the idea here is to leverage the capability of the USV Saildrone to go wherever you want it to go and not just follow the ship track. Thus, its sampling pattern does not have to look like the ship track and that is even why we applied it here. In addition, according to a Science Magazine News by Paul Voosen (March 8, 2018), nUSV can be easily driven remotely to cover desired sampling locations or patterns, plus the fact that it has a much higher sampling frequency (hourly).*

*Regarding the number of nUSVs it would take to sample the full Southern Ocean with a similar density, we are planning to provide an estimate in a follow-up study covering the Southern Ocean as a whole. The idea is to figure out how the carbon cycle community can better use existing and newly developed autonomous sampling vehicles to better supplement ship-based observations that are essential in machine-learning-based mapping approaches. However, longitudinally, the Southern Ocean is equal to 360°/20° = 18 times our 20°W-E domain. This means that it would take 18 Saildrones to sample the full Southern Ocean at that density (cf. Fig.1c, manuscript).*

**- The goal of the floats is to give the large spatial structure and this aspect of them is not valued in your experiment design. This should be mentioned when giving recommendations.**

*Response: Thank you for this suggestion. Contrary, in order to not disadvantage the floats, we deployed one float in each sub-domain (SAZ and PFZ), under the assumption that floats would not cross the Sub-Antarctic Front (SAF). In addition, given the size of the chosen domain and sampling frequency of the floats, we assumed that in this experiment design a larger spatial structure would be less realistic with a single float because in reality, the float stays at the surface only for a short amount of time. However, we do acknowledge that might create some biases, and that is why we mentioned that floats would benefit from more coordinated deployments to be able to resolve variability scales such as the meridional gradients. In the revised manuscript, when given recommendations, we emphasize this aspect more.*

**- If one were to extrapolate this to the full Southern Ocean, which is about 100 times larger than your box, what would the monetary cost to sample with the density of your experiment for each platform? For example, if it would take 100 nUSVs or 200 wavegliders is that practical?**

*Response: We thank the reviewer for this question. Here, we recall that the study is more about the mode of variability. The chosen domain is known to be a high-EKE area and based on the seasonal cycle reproducibility (SCR) of model pCO2 (Fig. 2a), it is representative of the model of variability we aimed to address in the Southern Ocean.*

*Further, as we mentioned previously, the whole idea is to figure out how the carbon cycle community can better benefit from autonomous sampling robots in order to supplement ship-based observations. From this study, we proposed that the meridional sampling approach may be an efficient way of sampling using autonomous robots. Therefore, in a follow-up study, we are carrying out similar experiments to the full Southern Ocean and also investigating how many nUSVs will take into practice to supplement ship-based observations in order to resolve both the intra-seasonal variability and the meridional gradients in the Southern Ocean as a whole.*

*According to a Science Magazine News by Paul Voosen (March 8, 2018), however, Saildrone Inc charges about $2500 a day per nUSV to collect ocean data, whereas ship time can cost $30,000 or more per day. Thus, based on the Sutton et al. (2021)-USV example (Fig. 3, of this note) that would sample hourly the study domain in about 16 days, to sample the full Southern Ocean with the density our experiment would cost about 18\*16 \* $2 500 = $720 000.*

**- Even in your box you may give some information about the monetary cost of each sampling effort to give some practical perspective.**

*Response: Thank you for the comment. The revised manuscript takes that into account based on the discussion about the previous point.*

**- WGs can be "driven", but you have chosen to give them a mooring-like program. This impacts the results. You finally mention this at the end, but not in the abstract when giving recommendations.**

*Response: Indeed, WGs were driven to follow a hexagonal pattern of about ±10km of radius as in practice (cf. Fig. S1 and Monteiro et al., 2015). However, WGs are harder to drive remotely (especially in the presence of jets), and relative to nUSVs they cannot cover a lot of space. Actually, the main reason for the pseudo-mooring mode was because of the assumption in the experimental design that was - intra-seasonal modes are critical to address uncertainties and biases. At that stage, there was little recognition of the seasonal variability of the meridional gradient.*

*In addition, realistically, the research vessel SA Agulhas, for example, crosses the study region at the same points making WGs deployment and retrieval a lot easier when they are done around the same spot. Therefore, from a logistic perspective, WGs were given a mooring-like program whereas nUSV would be able to sail to the next port.*

**- You give the caveat "We emphasize that these experiments are intentionally made to reproduce the sampling resolution of their real-world counterparts, not necessarily their spatial resolution in practice but at least the temporal one." In the text, but it needs to be featured more prominently (e.g. in the title).**

*Response: Thank you for suggesting this. We have reworded the sentence and made it feature more prominently in the abstract.*

**- It appears you train with mixed layer depth (MLD) for all platforms, but surface platforms don't give MLD. How will you get that information and deal with the uncertainty from this? This wasn't discussed, or I missed it, and this is a serious caveat.**

*Response: We thank the reviewer for this question. We do acknowledge that surface platforms do not provide information about the mixed layer depth (MLD). In practice, for instance, existing reconstruction methods use MLD climatology as a proxy variable (Gregor et al., 2019; Gloege et al., 2021). The main purpose of taking MLD climatology is to smooth the data and thus reduce the uncertainty from MLD information. Our approach of using MLD from the model rather than a climatology is an advantage compared to the existing methods (SOM-FFN, CSIR-ML6, etc.) that use climatology. However, we do acknowledge that this is a serious caveat and think it could be addressed in another study.*

**- Temporal aliasing by the floats is exacerbated by the fact you don't advect them in the model, so their quasi-Lagrangian nature is not taken advantage of. If you plotted pCO2 in figure 9 following a water parcel (or plotted it by integrating the full material derivative of pCO2) I assume it would be much smoother. What you show in figure 9 is likely dominated by advection and not air-sea fluxes. This discrepancy impacts your findings and should be addressed.**

*Response: Thank you for this comment. Actually, just in case the reviewer missed it, in lines 245 and 765-768 of the manuscript, we did mention that to mimic the quasi-Lagrangian nature of the floats,*

*we designed the floats to follow an eastward (water direction) Brownian motion or random walk (line 246) in order to add an element of stochasticity that is likely close to reality (lines 765-768).*

*Further, we look at the difference between the distance a float would cover over a 10-day period the surface (0 m) and at 1000 m (Fig. 4a-d, of this note). It shows that the distance the float would cover zonally (Fig. 4a-b, of this note) or meridionally (Fig. 4c-d, of this note) at the surface and deep layers would not make a big difference with respect to the variability that the model has over the 10-day period. It also appears that the offset between the two layers is very significant. This demonstrates that temporal aliasing by the floats is not as exacerbated as the reviewer thought. Nevertheless, we do acknowledge that this might impact our findings, and we would have addressed it more clearly.*

[Figure]

***Figure 4:*** *Estimates of the zonal (first row) and meridional (second row) distances traveled over a 10-day period (of the float) by two points (one in the SAZ and the other in the PFZ) at the surface layer (0m) and deep layer (1000m).*

**- Related to the figure 9 comment above, it seems odd to conclude the sampling need be at least daily (line 793) and that the floats should sample 1-2 days when the model output used is daily averaged. How much signal is really being aliased? Please do a more thorough time series analysis (e.g. show power spectra density to see how much variance is truly being aliased). You call into question a discrepancy with Bushinsky et al 2019. I suggest you repeat the calculation done in that paper with the model output. Since you have the model time series I suggest you be quantitative with your statements.**

*Response: We thank the reviewer for this comment. Actually, this study proposes that one would need more coordinated deployments of floats to resolve, for example, the intra-seasonal variability that nUSV and WGs were able to resolve. We think this synoptic-scale was resolved because of the high sampling frequency of nUSV and WGs, which would mean that sampling surface ocean CO2 at the correct interval remains critical. That is why we proposed the 1-2 day sampling period. In addition, based on the sampling period sensitivity analysis (cf. Fig. 5), Monteir et al. (2015) showed that to achieve the 10% uncertainty threshold in the Southern Ocean as discussed in Lenton et al. (2006), a sampling period of 1-3 days is necessary in areas of high EKE and elevated sub-seasonal dynamics like that of our study domain. Thus, having a perfect knowledge of how we need to sample, one can get an estimate of the sensitivity of pCO2 to various sampling frequencies.*

[Figure]

***Figure 5:*** *A map from Monteiro et al. (2015) showing a mean for the adaptive sampling interval (in days) required to achieve the 10% uncertainty threshold in the Southern Ocean as discussed in Lenton et al. (2006) .*

**- My other major criticism is that one expects mapping estimates to never get worse when more data is added. If they are getting worse when more data is added it signifies a flaw in the mapping method and not the data sampling itself. The fact that SHIP(smr) + FLOAT(SAZ+PFZ) has a significantly higher overall RMSE than SHIP + FLOAT(PFZ) is troubling. It points to either a serious issue with your methods or that the uncertainty in the RMSEs you are discussing are very large. You need to explain what is going on here. (This also calls into question your claim of the importance of temporal sampling as it appears something else is going on as well.) Did you carry out WG(SAZ+PFZ), and if so did it also have this troubling behavior?**

*Response: We thank the reviewer for this comment. Actually, the float deployed in the PFZ does very well, see  SHIP + FLOAT(PFZ), but when it is deployed in the SAZ it performs poorly, see SHIP + FLOAT(PFZ). We think this discrepancy results from the difference in the modes of variability of the SAZ and PFZ. Therefore, when we add together samples from these two regions (SAZ and PFZ), the resulting RMSE and bias get worse.*

*On the other hand, we ran the experiment several times, and the fact that SHIP + FLOAT(PFZ) kept outperforming SHIP + FLOAT(SAZ+PFZ) could also be explained as follows. When you have more samples in a certain region, you are more likely to cause the trained model to weight certain observations more. That weighting would be reflected in the outputs. We are not saying this is the right answer but it is also a possibility. In addition, the fact that SHIP + FLOAT(SAZ+PFZ) outperformed SHIP + FLOAT(SAZ) suggests that though there is more data, having regionally imbalanced samples would have also caused a significantly lower overall RMSE than SHIP + FLOAT(PFZ). We do acknowledge that one would expect mapping estimates to get better when more data is added, and when the opposite happens the first instinct is usually to suspect a flaw in the mapping method and not the data.*

*Regarding the last point, we did not carry out SHIP + WG(SAZ+PFZ) because the idea of carrying out SHIP + FLOAT(SAZ+PFZ) was to investigate whether we could take advantage of the larger spatial structure of the floats to resolve the meridional gradients.*

**Minor comments:**

**- Model is 1 yr, but the Southern Ocean has significant interannual variability and this should be discussed a bit more.**

*Response: We thank the reviewer for this suggestion. We have added sentences to further discuss the use of one year of high-resolution model data in the Southern Ocean despite the significance of the interannual variability in the region. However, we recall that the study deals with sampling scale problems stemming from synoptic-scale or intra-seasonal variability and the meridional gradients.*

**- Line 121 sentence is incomplete**

*Response: Thank you for pointing that out. We completed that sentence in the revised manuscript.*

**- Supplementary Information text needs cleaning.**

*Response: Thank you for this comment. We have cleaned the Supplementary Information text as suggested by the reviewer.*

**- In SI you say what hyper-parameters you tune but you don't give the final values you used. You should give these for reproducibility and to inform the results.**

*Response: We thank the reviewer for this comment. Indeed, the model hyper-parameters were tuned using K-fold cross-validation (CV) incorporated in the model training. This tuning was achieved using Bayesian optimization. The optimal values of hyper-parameters used were reported at the end of the model training and are now included in the revised Supplementary Information for reproducibility.*

**- Some typos in figure captions. Check these.**

*Response: Thank you. We have checked the typos in figure captions as suggested by the reviewer.*

**- Time, J, is stated as a predictor in some cases and not in others. Please check for consistency.**

*Response: Thank you for this comment. We have revised the manuscript to check for consistency in the usage of time/J as a predictor variable.*

---

## Author Comment (AC2)

*Response to reviewer 2 of "**The sensitivity of pCO2 reconstructions in the Southern Ocean to sampling scales: a semi-idealized model sampling and reconstruction Approach**"*
*by Djeutchouang, Chang, Gregor, Vichi, Monteiro*

*First of all, we would like to take the opportunity to thank you for the thoughtful comments and suggestions. Meanwhile, we have revised the title of the study as follows: "**A semi-idealized model sampling and reconstruction approach across a Southern Ocean front: the sensitivity of pCO2 reconstructions to sampling scales**". We will respond (in italics) to each of your specific comments as follows.*

**Summary and overall impression**

**The paper addresses an important question for the global carbon cycle community: how to reduce the uncertainties and biases of machine-learning-based mapping approaches in the Southern Ocean, a data-sparse but globally important region. The authors create synthetic data by subsampling a high-resolution model in a subregion of the Southern Ocean over 1 year. The synthetic data resembles different observational platforms in terms of the typical temporal resolution of the different platforms. These platforms include ship, float, Windglider, and Saildrone data. They then run two different machine learning mapping approaches with these synthetic observations and compare the mapped reconstruction of the seasonal cycle to the actual model field to estimate the biases and uncertainties. They run the method multiple times with different subsets of synthetic data to highlight how sampling in different seasons, as well as with different types of observational platforms affect the uncertainty and bias. They find that the addition of wintertime ship data would greatly reduce the errors in the reconstructions. They also find that Saildrones are an optimal platform to address both the large-scale spatial and high-resolution temporal sampling and have the most effective impact on reducing the uncertainties and biases of the seasonal and annual mean reconstructions of air-sea CO2 fluxes in the Southern Ocean.**

**This paper addresses a crucial gap in our current knowledge and provides suggestions on how the carbon cycle community can improve current estimates of the Southern Ocean carbon fluxes, through an improved sampling strategy. I very much support the method of using synthetic data to create a sampling strategy, the paper is well-written and has clear figures that support the findings. However, I have some major comments, which I believe should be addressed before publication. In my review, I mostly focus on the Methods section, as requested by the editor.**

*Thank you.*

**General comments**

**• My main concern is that the machine-learning approach used to reconstruct the model fields is very different to the common methods (e.g., by Landschuetzer, and Gregor…) and thus, I am not convinced that the lessons learned from the authors' approach can be directly translated to these methods. Specifically, the established mapping methods use training data from quite large regions (from a clustering step), which are a lot bigger than the region of this study. Thus, more**

**data flows in, and they might be more robust to be able to reconstruct the seasonal cycle despite the sparsity in winter data. In addition, there are zonal differences and hot spots of in the Southern Ocean, and the subregion might not be representative for the Southern Ocean as a. whole. I do think we can still learn from this current study, but this issue should be discussed thoroughly. A follow-up study could later focus on the Southern Ocean as a whole (or even globally within e.g., the clusters by Landschuezter or Gregor et al.).**

*Response: Thank you for taking the time to provide these important comments. We did look into the two commonly-used reconstruction methods by Landschutzer et al. (2014) and Gregor et al. (2019) that both adopt a two-step machine learning (ML) approach in which the first step consists of clustering the reconstruction domain whereas the second step applies ML regression and mapping in each cluster generated. We are aware of the necessity of this clustering step to overcome the spatial and temporal limitations of observations. In Fig. (1a), we illustrated the Southern Ocean Fay and McKinley (2014) biomes, one of the clustering methods used by Gregor et al. (2019) This figure helps to understand the motive of skipping the clustering step in this particular study as it shows that the clustering step was not necessary given the size of the study domain.*

*This study domain (black box, Fig. 1a, of this note) was not only spatially and temporally coherent but also big enough to reflect the spatial and temporal variability necessary to provide sufficient sensitivity to the different sampling strategies. We also recall, as we did with reviewer 1, that while our selected domain does not resolve all Southern Ocean scales, it is representative of the scale variability we aimed to address. This sub-region is roughly 50% STSS/SPSS of the Southern Ocean (Fig. 1a-b, of this note). Further, it is divided by the Sub-Antarctic Front (SAF) and thus also overlaps both the Sub-Antarctic Zone (SAZ) and Polar Frontal Zone (PFZ) which are relatively the two most sampled regions of the Southern Ocean.*

*We do acknowledge that the sub-region is smaller compared to the clusters and that might have some impacts. That is why a follow-up study is being conducted to extrapolate and test the idea in the entire Southern Ocean while including in the method a clustering step like in Landschutzer et al. (2014) and Gregor et al. (2019) in order to overcome the spatial and temporal limitations of observations.*

[Figure]

**Figure 1:** *Panel **(a)** is the Southern Ocean regions or biomes (Fay and McKinley, 2014) as extended and used in Gregor et al. (2019) on which are added the Sub-Antarctic Front (SAF) (red line) and the study domain (black box); and panel **(b)** show the fraction coverage estimates (%) of the two most sampled regions: STSS and SPSS biomes relatively to the area of our box. EB biome stands for Eastern Boundaries (Gregor et al., 2019). For other biome abbreviations (below the colour bar), see Fay and McKinley (2014).*

**• I think it's great that the authors use an ensemble of two ML-based approaches. However, I would appreciate a short analysis of how the two estimates differ, to understand how robust the findings are.**

*Response: Thank you for this comment. Indeed, we reported in the Supplementary Materials (Table S3) a short analysis of the in-sample errors of the two members of the ensemble for all the different sets of experiments we performed.*

**• The authors say that there is very limited data to allow for both training and testing/validation data. But how do the authors then know that the outcome is not overfitted? As the data is synthetic, could one not add more synthetic data that would then allow for both training and testing data?**

*Response: Thank you for these questions. Indeed, given the size of the study region and the idea of mimicking as much as possible the sampling scales of observing platforms, the sample sizes were relatively small to afford splitting the simulated observations for both training and testing.*

*However, to answer the question about overfitting was addressed as follows. To better control the overfitting, we incorporate a K-fold cross-validation (CV) during training in order to find the set of hyper-parameters that enable a better generalisation of ML2. The K-fold CV is applied identically to each of the two-member algorithms (like in Gregor et al., 2019) and the tuning of hyper-parameters was achieved using Bayesian optimization instead of the standard grid-search CV. The optimal values of hyper-parameters used were reported at the end of the model training and are included in the revised Supplementary Information for reproducibility.*

*About the question regarding adding more synthetic data to allow both training and testing data, in fact, that is what we are doing by comparing our results with the model data (known truth) that were not involved in the synthetic platforms simulations.*

**Specific and minor comments to the text:**

**Introduction:**

**• I found the introduction a bit misleading. After reading the introduction, I expected that the paper would include the interannual to decadal variability, but it "only" focuses on the seasonal cycle based on data from one year. Consider rephrasing this to not disappoint the reader.**

*Response: Thank you for this suggestion. We have carefully taken this into consideration and the revised introduction refocuses on the seasonal cycle based on one year-round data in a semi-idealized subdomain of the model within the Southern Ocean. It also emphasizes the two most observed regions of the Southern Ocean; that is, the Sub-Antarctic Zone (SAZ) and Polar Frontal Zone (PFZ) whose the chosen subdomain represents the most.*

• **Similarly, the introduction should mention clearly that this study "only" focuses on a subregion within the Southern Ocean.**

*Response: Thank you for the suggestion. The revised introduction takes it into consideration.*

• **Gloege et al. 2021 did an in-depth analysis of the uncertainty of ML-based mapping approaches, using synthetic data at a global scale. I think the introduction should mention that study and be explicit about how this current approach differs and what's new about this study in comparison.**

*Response: Thank you for this suggestion. We are aware of the Gloege et al. (2021) study and the advances their work made in this area of research. The revised introduction makes a deeper and explicit connection with that study and explicitly mentions how our approach differs.*

• **L.114: It's mentioned later that how well the model matches the observations does not really matter in this context. However, please consider mentioning here already why using that model works (considering e.g., the Mongwe et al. 2018 study that showed how the CMIP models completely disagree on the phase and magnitude of the seasonal cycle).**

*Response: Thank you. Regarding this comment, the revised version takes it into consideration by giving further explanation around that statement as follows. Having a complete model data, the full domain-truth knowledge of pCO2 variability can be assumed to be known independently of that particular model constraints. Therefore,we think using any physics-biogeochemistry forced ocean model output (including any model from CMIP5 models) would work as long as the model can represent the range of temporal and spatial modes of variability that are necessary to mimic the sampling scales of interest.*

*The modes of variability that the forced model (NEMO-PISCES) captures are still responses of pCO2 that are in some way related to the changes in driver variables (or related proxies). Thus, for the purpose of this study, the "correctness" of the pCO2 response to the driver variable is not the most important. What we try to show is that the reconstruction is sensitive to the way one samples with different modes of variability.*

• **L.196: Is this really the case? I would assume that any differences between the model and the observations could matter. I.e., the model might be generally a lot smoother than reality and thus the sampling strategy might be less sensitive in the model than in the real world. I would appreciate a short discussion on that.**

*Response: Thank you for this question and suggestion. Actually, the forced coupled ocean model (NEMO-PISCES) used in the study does not represent the full processes driving the CO2 in the Southern Ocean. As for the machine learning (ML) methods used in this study; that is, the feed-forward neural network, and the tree-based method gradient boosting machine, they do not capture the mechanisms that actually drive the pCO2 (Holder and Gnanadesikan, 2021). Rather, these ML methods capture changes in the drivers and then the associated changes in the pCO2.*

*Therefore, we thank the reviewers for their suggestion of assuming that any differences between the model and the observations could matter because this could mean the pCO2 may be driven by a mechanism or process different from the real world. Indeed, this does seem like a plausible option given that model pCO2 is largely driven by the SST as in the real world.*

• **L.335: I think the explanation of error and uncertainty is the wrong way round.**

*Response: Thank you for this comment. We took it into consideration by revising our explanation of the two concepts as follows. "The pCO2 total uncertainty (E) is dealt with as in Gregor and Gruber (2021). The authors identified within the surface ocean carbonate system three main sources of errors that contribute to E. This included (1) the measurement (M), (2) representation (R), and (3) prediction (P) errors. Under the assumption that these three components are independent of each other in the pCO2 uncertainty space, E can thus equivalently be expressed as …"*

---

## Author Response (AR2)

Rebuttal to "**The sensitivity of pCO2 reconstructions in the Southern Ocean to sampling scales: a semi-idealized model sampling and reconstruction Approach**"

by Djeutchouang, Chang, Gregor, Vichi, Monteiro

**Overview of the changes**

First of all, we would like to take the opportunity to thank the reviewer for the thoughtful comments and suggestions to strengthen and improve the manuscript. We have made a few changes to the manuscript, which we hope address those comments. Two main changes are (1) the greater emphasis on the caveats of including in our experiment the mixed layer depth and surface chlorophyll-a from the model; and (2) an additional quantitative analysis of the implications of the 10-day sampling period y.

In the document below, we show the response of the reviewer in bold and blue, and the response to each point in black/italics. The track changes are shown in a mark-up version of our revised manuscript in a separate file according to the "review file upload" guidelines.

**Overview**

**The authors did an excellent job addressing my concerns. The paper is improved. There are just two concerns left that I am not satisfied with.**

*Thank you.*

- **The first is regarding the MLD caveat. This is not made clear enough. A well-placed clear sentence about this issue should suffice. Otherwise, you either shouldn't use MLD in the reconstruction or only predict using accurate MLD from the float data. The latter would be an interesting experiment. This criticism is also relevant to some extent to chlorophyll, as surface chlorophyll is not the same as what the satellite measures and depth resolution can also be important for this variable.**

*We thank the reviewer for this comment. Studies (such as Gregor et al., 2019; Devil-Sommer et al., 2019; Gloege et al., 2021) have found that MLD is an important driver, and so we wanted to keep the MLD as the regression metrics also pointed to a significant contribution by the MLD. However, we do recognize that using the model MLD in the reconstruction is relatively an advantage as we mentioned in the manuscript (Lines 143-45). We also acknowledge that the surface chlorophyll in the model is not the same as the surface chlorophyll that satellites measure. More specifically, the advantage of*

*including both proxy variables is that the model is providing constraints which may not be available from real-world observations. However, these advantages are relative to real-world observations, and uniform across all the sampling experiments in this study.*

- **My other remaining criticism is that I still am concerned about my previous comment: "Related to the figure 9 comment above, it seems odd to conclude the sampling need be at least daily (line 793) and that the floats should sample 1-2 days when the model output used is daily averaged. How much signal is really being aliased? Please do a more thorough time series analysis (e.g. show power spectra density to see how much variance is truly being aliased). You call into question a discrepancy with Bushinsky et al 2019. I suggest you repeat the calculation done in that paper with the model output. Since you have the model time series I suggest you be quantitative with your statements." This is important given that the abstract concludes "Wavegliders with hourly/daily resolution in pseudo-mooring mode improve on Carbon-floats (10-day period), which suggests that sampling aliases from the 10-day sampling period might have a greater negative impact on their uncertainties, biases, and reconstruction means;"**

*Thank you to the reviewer for making this suggestion. We acknowledge that there is a limitation with the model output as its fine-scale resolution might not be high enough. However, using hourly observations, Monteiro et al. (2015) did show that 1-day sampling frequency was sufficient to capture the variability so that uncertainty was minimized within a 10% uncertainty threshold in zones characterized by $CO_2$ intra-seasonal dynamics such as the SAZ. This justifies why the 1-day model output was used and why we deem it sufficient. However, increasing the model frequency from daily to hourly would likely reveal that the uncertainty of sampling sporadically at a 10-day resolution increases uncertainty even more due to large dynamic variability and diurnal variability (Torres et al. 2021; Monteiro et al. 2015).*

- **Perhaps the answer with regards to a discrepancy with Bushinsky et al 2019 is related to your finding that "The float did well when deployed in PFZ dominated by seasonal variability which can be resolved by the 10-day sampling period but performed poorly when it was deployed in the SAZ characterized by intra-seasonal modes which cannot be resolved by the 10-day sampling period." It is possible that the discrepancy is due to location, as I believe Bushinsky et al used data from the SOFS mooring. There are many ways to be more quantitative about this issue. One suggestion is to filter (i.e. band-pass) the model output and show where in the domain pCO2 variability at frequencies higher than 10 days is a significant fraction of the total variability. This straightforward calculation is necessary to support the conclusions being drawn.**

*We thank the reviewer for this good suggestion, which we implemented below. While we have not directly applied a low band-pass filter, we used instead a 10-day rolling mean, which in this instance may be a simpler and more transparent way to achieve the objective of the low band-pass filter.*

*The aim of the 10-day rolling mean is to eliminate or weaken all the frequencies higher than the 10-day mode of variability. We took this into consideration in the revised version of the manuscript.*

*Specifically, in order to be more comprehensive while providing a more quantitative characterization of our findings, an additional analysis was conducted on the 10-day modes of variability. We thus took the difference of this 10-day rolling mean from the daily model output, which would give us the < 10-day modes of variability while its root-mean-squared error (RMSE) would give us a statistical understanding of what the uncertainty might be if we sampled at a 10-day rate. This approach is actually similar to what Mazloff et al. (2018) did in their study but they applied a 90-day rolling mean.*

*In this approach, the hypothesis is that by weakening the < 10-day modes of variability, we should be able to show a significant improvement in the RMSE for the 10-day sampling mode. Our analysis resulted in an average of 2.53 µatm in the SAZ and 1.71 µatm in the PFZ. The study domain map of the RMSEs is shown in Fig. 1 (on this note) which also shows a significantly high uncertainty in the SAZ.*

[Figure]

**Figure 1:** *The RMSE map of the difference of the 10-day rolling mean (i.e., low-pass filtered pCO2 where the duration is set to 10 days) from the daily model pCO2 in the study domain divided by the Sub-Antarctic Front (SAF, red dashed line) into two sub-domains: the Sub-Antarctic Zone (SAZ) and the Polar Frontal Zone (PFZ).*

*These results show a dramatic decrease in the RMSE relative to the 1-day analysis, highlighting thus the important contribution made by the < 10-day modes of variability from the daily model output. This map (Fig. 1, on this note) confirms the sensitivity of the RMSE of 10-day sampling reconstruction to the presence or absence of synoptic variability, which was also highlighted in Monteiro et al. (2015).*

---

## Author Response (AR3)

Rebuttal to "**The sensitivity of pCO2 reconstructions in the Southern Ocean to sampling scales: a semi-idealized model sampling and reconstruction Approach**"

by Djeutchouang, Chang, Gregor, Vichi, Monteiro

**Overview of the changes**

First of all, we would like to take the opportunity to thank the Associate Editor's decision to publish our manuscript subject to technical corrections to suit the BG guidelines. We have made a few technical changes to the manuscript figures as recommended.

In the document below, we show the response of the Associate Editor in bold and blue, and the response to each point in black/italics.

**Overview**

**Dear authors,**

**Many thanks for revising your manuscript. I am happy to accept it in its present form but would like you to rework the colour scheme used in Figures 4 and 8 before uploading the figures for final publication. The BG guidelines specify that "it is important that the colour schemes used in your maps and charts allow readers with colour vision deficiencies to correctly interpret your findings. Please check your figures using the Coblis – Color Blindness Simulator and revise the colour schemes accordingly." The combination of red and green, particularly with the red dashed line in Figure 4 might cause difficulties for some readers.**

**Best regards**
**Peter Landschützer**

*We thank the Associate Editor for the recommendation. Following the BG guidelines for colour schemes used in the maps and charts, we revised the colour schemes we used in Figures 4a and 8b of the final manuscript. We chose a more perceptually uniform colour scheme, "RdYlBu", which is different from "jet". After testing the resulting maps through the Coblis-Color Blindness Simulator, they now look much more colour-vision-deficiencies friendly compared to using the "turbo" colour scheme that was initially used. The red dashed line in Figure 4a has been changed to a black dashed line.*

*We thus took this into account before uploading the files required for the production process.*